

**A database of radiogenic Sr-Nd isotopes at the "three poles"**
Zhiheng Du[a], Jiao Yang[a], Lei Wang[b], Ninglian Wang[c], Anders Svensson[d], Zhen
Zhang[e], Xiangyu Ma[f], Yaping Liu[a], Shimeng Wang[a], Jianzhong Xu[a], Cunde Xiao[b*]
[a]State  Key  Laboratory  of  Cryospheric  Science,  Northwest  Institute  of
Eco-Environment and Resources, Chinese Academy of Sciences, Lanzhou 730000,
China
[b]State Key Laboratory of Land Surface Processes and Resource Ecology, Beijing
Normal University, Beijing 100875, China
[c]College of Urban and Environmental Sciences, Northwest University, Xi'an 710127,
China
[d]Centre for Ice and Climate, Niels Bohr Institute, University of Copenhagen,
Copenhagen, Denmark
[e]School of Spatial Informatics and Geomatics Engineering, Anhui University of
Science and Technology, Huainan 232001, China
[f]Qingdao Blue Thinking Information Technology Co.Ltd, Qinggao 266555, China
*Correspondence  and  requests  for  materials  should  be  addressed  to  CD. X
(cdxiao@bnu.edu.cn).
**Abstract:** The radiogenic isotope compositions of strontium (Sr) and neodymium (Nd)
on the surface of the Earth are powerful tools for tracing dust sources and sinks on
Earth's surface. To differentiate between the spatial variabilities of aeolian dust
sources in key cryospheric regions at the three poles (including the 'Third Pole'
covering the high mountainous area in Asia, the Arctic and Antarctica), a dataset of
the Sr-Nd isotopic compositions from the terrestrial extremely cold or arid
environments in this study was compiled, similar to the method of Blanchet (2019).
The database identified snow, ice, sand, soil (loess) and sediment from the modern





dust samples and paleoclimatic records of the three poles based on 43 different
references, with a total of 967 data points. There are 274 data points from the third
pole, 302 data points from the Arctic, and 391 data points from Antarctica. The
sampling and measurement methods and the quality of these data are recognized and
introduced. For each pole, geographical coordinates and other information are
provided. The main scientific purpose of this dataset is to provide our own
measurements and collect documentation for the Sr-Nd dataset, which will be useful
for determining the sources and transport pathways of dust at the three poles and to
investigate whether multiple dust sources are present at each of the poles. This dataset
provides exhaustive detailed documentation of the isotopic signatures at the three
poles during specific time intervals, which are useful for understanding the dust
sources or sinks of the three poles. The datasets are available from the National
Tibetan Plateau Data Center (https://doi.org/10.11888/Cryos.tpdc.272100, Du et al.,

2022).

**Keywords**: Radiogenic isotopic dataset, third pole, Arctic Ocean, Greenland and
Antarctica ice sheets, Dust provenances.
**1. Introduction**
The role of mineral dust in the Earth system extends well beyond its impact on
the energy balance and involves the interactions with the carbon cycle, the cryosphere,
and public health on global scales (Shao et al., 2011). The transport of dust from the
low mid-latitudes, which contain major deserts that are dust sources, to the Arctic
region or Antarctic ice sheet is sensitive to amplified high-latitude climatic variability
(Lambert et al., 2013: Struve et al., 2020). The isotopic compositions of the
radiogenic isotopes of strontium (Sr) and neodymium (Nd) are a powerful tool for
tracing dust sources and sinks because their characteristics are significantly different


on the surface of the Earth (including sand, sediment, loess, aeolian deposits and snow)
(Grousset et al., 2005). Therefore, the combination of different isotopic signatures,
specifically $^{87}Sr/^{86}Sr$ and $^{143}Nd/^{144}Nd$, has proven be useful in discriminating different
dust source areas in the earth science (Biscaye et al., 1997).
Aeolian dust from East Asian deserts is transported globally and has been found
in Greenland snow and ice based on Sr-Nd data in modern environments (Bory et al.,
2003a; Bory et al., 2003b; Lupker et al., 2010). Sr-Nd data in the Greenland ice sheet
(GrIS) deep ice core also emphasize the contribution of aeolian dust from Asian
deserts (Biscaye et al., 1997; Svensson et al., 2000). The Sahara was taken as an
additional dust source for the GrIS based on NEEM and Dye-3 ice cores (Han et al.,
2018; Lupker et al., 2010). This situation has been attributed to the lack of detailed
observational data in the remote Arctic; therefore, the prediction ability of the impact
of changing aeolian dust loading on cryospheric science has been limited in the
Northern Hemisphere (NH). Therefore, aeolian dust from various source regions,
including the Saharan Desert in North Africa or the Gobi and Taklamakan Deserts in
Asia, is transported to the GrIS, there are still great uncertainties. As much more
Sr-Nd data were measured, and it is necessary to reassess these data on the dust
sources for the Arctic.
The transport of aeolian dust from natural desert regions has also been found in
modern snow and ice records at the third pole (Wu et al., 2010; Xu et al., 2012; Du et
al., 2015; Dong et al., 2018). Many studies have focused on dust transport from the
western Chinese deserts to the Chinese loess plateau (CLP), Pacific Ocean and even
the GrIS (Chen et al., 2007; Du et al., 2015; Wu et al., 2010; Wei et al., 2021).
However, it is still a controversial issue; for example, recent results have emphasized
that aeolian dust from local sources contributes significantly to high mountain glaciers



(Du et al., 2019a; Wei et al., 2021). On longer time scales, Sr isotope geochemistry
in fluvial sequences (12.7-4.8 Ma) reveals aeolian dust input and the southeastward
expansion of the dust impact on river water occurred on the northeastern TP (Ruan et
al., 2019). Therefore, the amounts of Sr-Nd data measured in snow, soil, sediment,
sand and other samples should be integrated into a dataset to better serve the
environmental and climatic sciences studying the third pole in the future.

The Sr and Nd isotope data from insoluble dust in snow samples along the

Zhongshan-Dome A transect, East Antarctica, indicate that long-distance natural dust
primarily originates from Australia and that local dust originates from ice-free areas
(Du et al., 2018). The Sr-Nd characteristics of snow layers at the Berkner Island ice
sheet in western Antarctica, for most of the year, are data support scenarios that
involve contributions from proximal sources (Bory et al., 2010). Sr-Nd in the Taylor
Glacier zero-age ice samples and snow samples from Roosevelt Island could be a
mixture of at least two local sources (Winton et al., 2016a; Aarons et al., 2017). The
Sr and Nd data from East Antarctica ice cores during the Holocene indicate a
well-mixed atmospheric background involving a mixture of two or more sources in
the SH (Aarons et al., 2016, 2017; Delmonte et al., 2019). Southern South American
(SSA) dust is considered to have been the dominant type of dust during glacial
periods in the Southern Hemisphere (SH) (Grousset et al., 1992). The amount of
isotopic information is currently adequate for Patagonian and non-Patagonian mineral
aerosols exported from southern South America (Gaiero et al., 2007; Delmonte et al.,
2010a, b, 2019; Aarons et al., 2017). However, because some ice-free areas are
located at the present-day margin of the East Antarctic ice sheet (EAIS), data are
insufficiently documented, and much is still known about the cycle in the SH. Major
efforts have attempted to solve the 'puzzle' of the origin of the potential source areas



that contribute dust to the Southern Ocean and the whole Antarctic ice sheet (Gili et
al., 2021). The Sr-Nd data in the entre Antarctica ice sheet have an uneven distribution.
Measuring Sr-Nd stable isotopic compositions in ice cores from Antarctica is a major
challenge. Therefore, these data characteristics and measurement methods are
discussed in detail.

The answers to these questions have been hindered by a paucity of Sr-Nd data,

which provide information on the local and potential dust sources. For these reasons,
we measured Sr-Nd data in some samples and collected Sr-Nd data in the literature at
the three poles (Fig. 1, Table 1). Therefore, the objective of this work was to produce
a compilation of published and unpublished data from the three poles, further discuss
the aeolian dust contributions from high-elevation regions, and trace the potential
aeolian dust transport paths in Greenland and Antarctic ice sheets. Similar to the
method of Blanchet (2019), here, we compile published and unpublished Sr-Nd data
with an integrated filtering system from three remote poles, in which these data were
collected in terrestrial extremely cold or arid environments with data augmentation,
and most of the data were not included in the previous dataset. The measurements of
Sr and Nd isotopic ratios used thermal ionization mass spectrometry for most samples.
The dataset will help trace modern natural dust, reconstruct past environments, and
extend the database of terrestrial radiogenic Sr and Nd isotopes in the Earth and
environmental sciences.
**2. Sample collecting, data measuring and processing**

Sr-Nd data in surface sand, soil or loess samples were collected from own

research and literature from three poles (including Chinese deserts and the Tibetan
Plateau (TP), the Greenland and Antarctic ice sheets, the Arctic, Australia, southern
South America (SSA), Southern Africa (SA) and New Zealand) (Fig. 1). The sand



(soil and loess) samples were collected from Chinese deserts and the TP, including the
Taklimakan Desert in the Tarim Basin, the Gurbantunggut Desert in the Junggar Basin,
the Qaidam Desert in the Qaidam Basin, and the Badain Jaran and Tengger Deserts, as
well as the TP and Chinese loess Plateau (CLP) (Fig. 2, Table 2). Sand or soil samples
in the Arctic were also collected from Ny-Ålesund on the western coast of Svalbard;
Barrow, Alaska; and Kangerlussuaq, West Greenland (Table 3). Cryoconite samples
were collected from the surface at different elevations in glaciers from western China
and GrIS (Table 3). The sand (soil, loess and other types) samples were sampled in
Australia, southern South America, Southern Africa and New Zealand, and
information can be found in Table 4. Four sand samples collected on King George
Island and eleven sand samples collected on Inexpressible Island in the Ross Sea,
West Antarctica, were measured in this study (Table 4). In general, the upper 2 or 5
cm of surface topsoil (sand) was collected with a trowel and stored in precleaned
plastic bags or bottles. Surface sediments from shelves and ridges in the Arctic Ocean
(Table 3), which were mostly retrieved from core archives, were subsampled in the
upper 10 cm of the core tops (with rare exceptions) (Maccali et al., 2018). Different
grain sizes (<5 μm, <10 μm, <71 μm, <75 μm and <100 μm fractions and bulk) of
surface soil or sand were extracted by the sieving method (Chen et al., 2007; Maccali
et al., 2018; Du et al., 2018, 2019a, b; Wei et al., 2021).
The snow samples were collected in the most favourable sector to avoid possible
pollution by camp activities (upwind from the camp according to prevailing summer
wind directions). Snow samples were collected from the snowpit at a vertical
resolution of 5–20 cm, following the clean-hands protocol with sampling personnel
wearing integral Tyvek® bodysuits, nonpowdered gloves and masks to avoid possible
contamination (Xu et al., 2012). In this study, one 1.0 m snowpit with a resolution of



10 cm was dug in the East Greenland ice sheet (GrIS), and four fresh snow samples
(M1, M2, M3 and M4) were sampled on sea ice in the Arctic Ocean during MOSAIC
(Multidisciplinary drifting Observatory for the Study of Arctic Climate) in October
2020 in this study. Surface fresh snow (2-10 cm) samples at different resolutions (with
different thicknesses, widths and lengths) in Greenland and Antarctica ice sheets were
excavated and placed in 5 L Whirl-Pak bags (Du et al., 2018; Du et al., 2019a, b).
Three horizontal snow layers were collected for Greenland and Antarctica snowpits
(Bory et al., 2003b; Bory et al., 2010). The dust from snow samples was extracted
using three methods. First, melt water was immediately filtered through a membrane
filter (with 0.2 or 0.45 μm pore sizes) by using precleaned (acid washed) plastic
filtration units (Wu et al., 2010). Second, melt water was centrifuged, with the unit of
at revolutions per minute (rpm), the supernatant was discarded and the remaining
water was vacuumed freeze-dried (or evaporated). The filtration was completed in a
class 1000 clean room (Xu et al., 2009). Third, the melt water was evaporated for
obtaining sufficient dust. The ice cores were kept frozen and sampled at different
intervals, which were drilled from the TP, Greenland and Antarctica ice sheets. Detail
of geographical coordinates and original information can be found in Tables 1-4 and
references. The dust in the ice core wass extracted using the same method as that for
the snow samples. Snow or ice core samples are nearly bulk samples or had different
grain sizes (>0.2 μm, > 0.45 μm, > 0.45 μm and <30 μm) (Du et al., 2015, 2019b;
Bory et al., 2003 a,b; Bory et al., 2010; Lupker et al., 2010; Wu et al., 2010). In
particular, the soluble fraction of some ice core samples was measured, which can
indicate marine or anthropogenic pollutant signals (Lupker et al., 2010; Du et al.,

2015).

Sr-Nd isotope datasets from snow, ice core, sand, sediment, soil and loess



samples from the third pole, Arctic and Antarctica were compiled. In total, 274 (snow,
ice, soil and sand (loess) samples) data points were collected from the third pole, 302
data points (snow, ice, cryoconite, sand, soil and sediment) were collected from the
Arctic (Table 3), and 391 data points (snow, ice, soil and sediment) were collected
from the SH and Antarctic ice sheet. The locations of these samples are shown on
maps provided below. To keep the naming scheme uniform, the dataset assembled the
names of each sample based on the work by Blanchet (2019). This dataset adds the
Sr-Nd isotope data from extreme cold and drought environments at the three poles
and built by incorporating data from the literature and our own database; in particular,
units and geographical coordinates are marked in the dataset. An overview of the
input data is shown in Table 1. The study focuses on the large amounts of different
data, including data on snow, ice, sand, soil, loess, sediment, etc. The data are based
on our own measurements, author contributions (data published and provided by first
author) and literature searches. Data were collected from 43 different references with
967 data points in total.
All subsequent procedures were performed in clean lab facilities. The sand, loess,
sediment, cryoconite and dust extracted from snow or ice cores were generally
digested with ultrapure acid ($HNO_3$, HF and $HClO_4$ or $HNO_3$, HF and HCl), and
$^{87}Sr/^{86}Sr$ and $^{144}Nd/^{146}Nd$ ratios were determined by the different types of
thermoionization mass spectrometry (TIMS). Sr-Nd values, with uncertainties are
expressed as $\pm 2\sigma \times 10^{-6}$ (2 standard errors of the mean) and also can be found in the
original references. The $^{144}Nd/^{146}Nd$ isotopic composition is expressed as:
$\varepsilon_{Nd}(0) = ((^{143}Nd/^{144}Nd)_{Sample}/(^{143}Nd/^{144}Nd)_{CHUR}-1)\times 10^4$, where $(^{143}Nd/^{144}Nd)_{CHUR}$
$= 0.512638$, where CHUR stands for chondritic uniform reservoir and represents a
present-day average Earth value $(^{143}Nd/^{144}Nd)_{CHUR} = 0.512638$ (Jacobsen &
Wasserburg 1980; Blanchet, 2019).

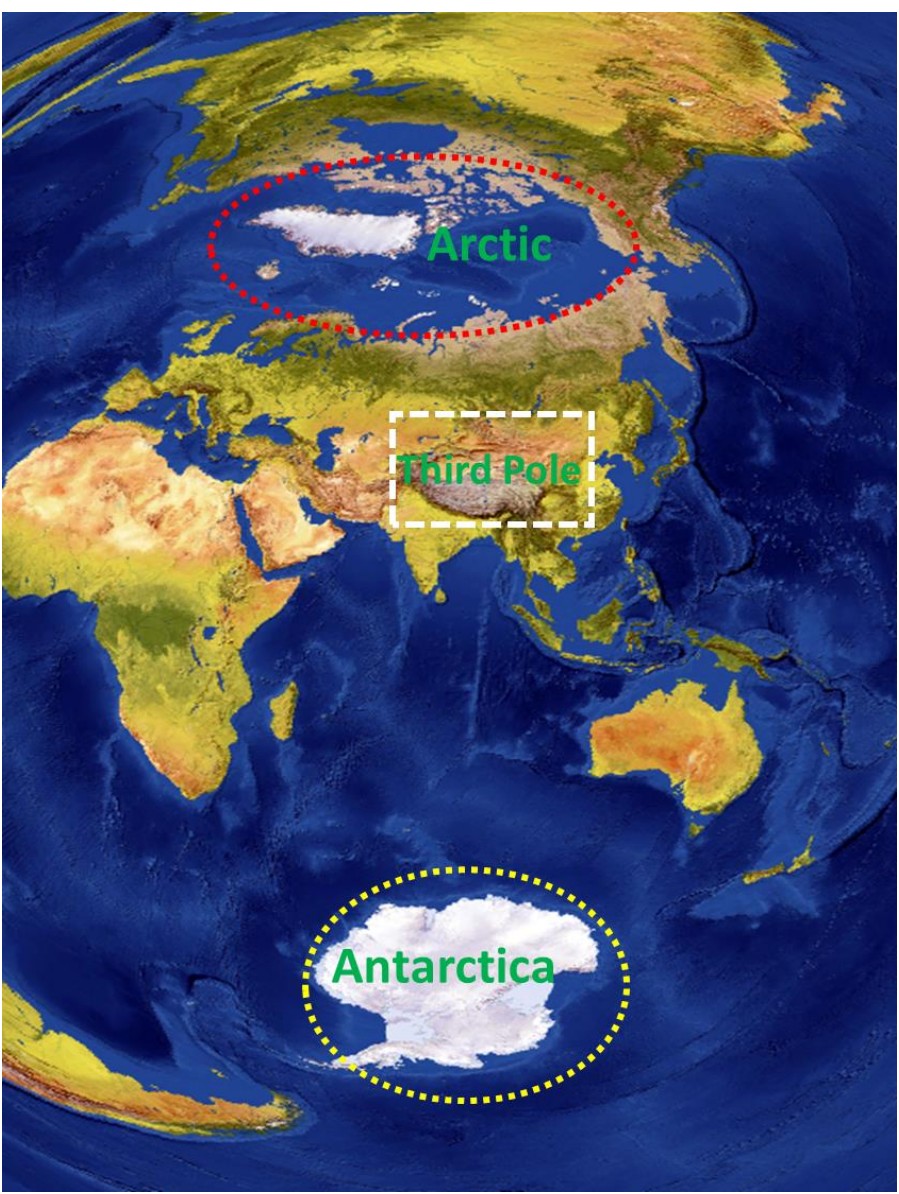


Fig. 1. Map of the sampling regions in the three poles (the third pole, Arctic and
Antarctica are indicated with different coloured circles) in this study (The background
of this figure is from ArcGIS).
**3. Data descriptions**





### 3.1. The Sr-Nd data measurement of glaciers at the third pole


Table 2 provides an overview of the information (the number of glaciers;
subregions; glacier name; site name: name of the sampling site where the samples
were taken; longitude and latitude; sample type and elevation) from the third pole,
which are used to fingerprint potential source areas (PSAs) based on the isotopic
signatures of these snow and sand (soil) samples (Fig. 2; Table 2). The grain size
effect in different samples is presented in the dataset for better illustration using these
data. The grain size effect in different samples resulted in $^{87}Sr/^{86}Sr$ ratio variations
(Chen et al., 2007; Svensson et al., 2000; Gili et al., 2021). The different acid leaching
methods also have a weak effect on Sr isotopic composition in the silt and clay
fractions (Schettler et al., 2009; Naoko et al., 2010; Meyer et al., 2011). Therefore,
this work attempts to build a new database that includes the different gran sizes and
acid leaching methods.
This feature validates the use of the geological characteristics from Sr-Nd
isotopic data; as an example, we can use the sorting criteria for determining PSAs
based on these data. For introducing these data based on geographic features, six
isotopic subregions across the entire third pole were divided as follows (Fig. 3):
Region I: Samples from glaciers located in the Altai Mountains include the snow
samples from Musidao glacier and Altay, and sand samples from the Gurbantunggut
Desert, with $\varepsilon_{Nd}(0)$ values from -6.55 to -1.2 and $^{87}Sr/\ ^{86}Sr$ values ranging from
0.705483 to 0.71480. The highest $\varepsilon_{Nd}(0)$ values were observed in this region (Chen et
al., 2007; Xu et al., 2012; Du et al., 2019a).
Region II: Samples from the glaciers on the northern margin of the TP include
snow samples from the Tienshan Mountains (Tienshan No. 1 glacier and Miaoergou
ice cap) and Kunlun Mountains (Muztagata), as well as sand samples from the



Taklimakan Desert, with $\varepsilon_{Nd}(0)$ values from -11.8 to -6.9 and $^{87}Sr/^{86}Sr$ values from
0.70842 to 0.728641 (Chen et al., 2007; Nagatsuka et al., 2010; Du et al., 2015; Xu et
al., 2012; Wei et al., 2019).
Region III: The Sr-Nd isotopic characteristics of the glaciers and sand/soil in the
interior of the TP include $\varepsilon_{Nd}(0)$ values ranging from -10.5 to -8.6 and $^{87}Sr/\,^{86}Sr$ values
from 0.713192 to 0.721786 (Xu et al., 2012; Du et al., 2019a; Wei et al., 2021).
Region IV: The Sr-Nd isotope data from snow and sand (soil) samples from
glaciers in the Himalayan Mountains (East Rongbuk, Jiemayangzong and Yala)
include $\varepsilon_{Nd}(0)$ values ranging from -28.1 to -10.5 and $^{87}Sr/^{86}Sr$ values ranging from
0.724542 to 0.757407 (Xu et al., 2012; Wei et al., 2021).
Region V: Samples from the glaciers in the Qilian Mountains include snow
samples from the Qilian Mountains and sand (soil and loess) samples from the Hexi
Corridor, with $\varepsilon_{Nd}(0)$ values from 0.712349 to 0.73211 and $^{87}Sr/^{86}Sr$ values from -15.7
to -7.0 (Wei et al., 2017; Dong et al., 2018). The $\varepsilon_{Nd}(0)$ values have an increasing
trend along the Hexi Corridor from west to east: -15.7–-12.9 for Laohugou No. 12
glacier (local soil: -13.6), -13.7–-8.58 for Qiyi, -13.8–-13.6 for Shiyi glacier (local
soil: -13.8–-13.6), -12.1–-12.0 for Dabanshan snowpack, and -10.9–-7.0 for
Lenglongling glacier (Fig. 2, Dong et al., 2018). It is very clear that, based on local
soil data, regional dust makes a significant contribution to these glaciers.
Region VI: Samples from the glaciers in the eastern TP include snow and soil
samples from the Hengduan Mountains, with $\varepsilon_{Nd}(0)$ values from -17.1 to -10.1 and
$^{87}Sr/^{86}Sr$ values from 0.717145 to 0.735863 (Xu et al., 2012; Dong et al., 2018).
There is an increasing $^{87}Sr/^{86}Sr$ trend from north (region I) to south (region V),
and there is a decreasing $\varepsilon_{Nd}(0)$ trend from north (region I) to south (region V). The
maximum $^{87}Sr/^{86}Sr$ ratios and minimum $\varepsilon_{Nd}(0)$ values were observed in region V (Fig.



3). The Sr-Nd data in the third pole have relatively narrow ranges with distinct
features, while the largest uncertainty was observed from Region IV. The same
measurement methods were used in these references (Du et al., 2015, 2019a; Dong et
al., 2018, Wei et al., 2019, 2021), and a similar measurement method was used by Xu
et al. (2012). Different methods were used by the other two references (Chen et al.,
2007; Nagatsuka et al., 2010). However, the data results seem to remain fully
consistent with these references.

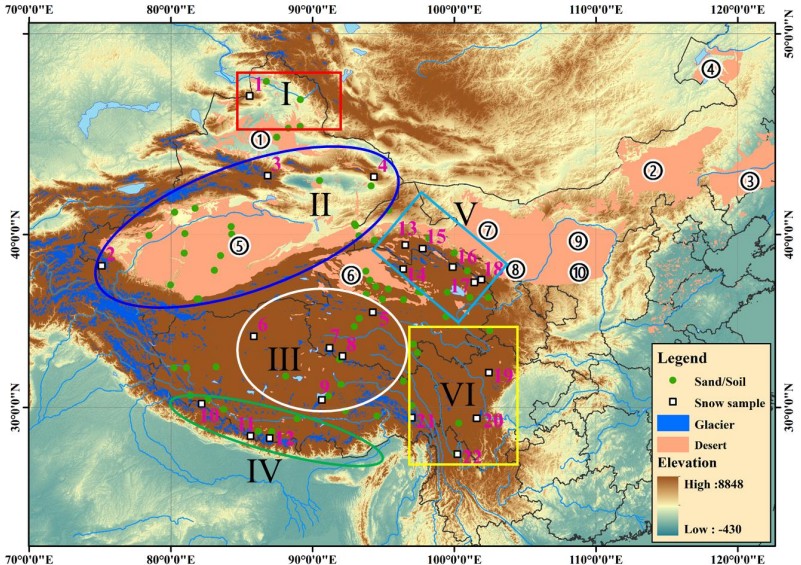


Fig. 2. The glacier and desert distributions in western China (the different coloured
oval and rectangular shapes represent six sub-regions based on Sr-Nd data; pink
numbers and white rectangles represent 22 glaciers, for which the names of glaciers
are shown in Table 2, and green solid circles represent sand/soil samples; the
numbered circles represent the ten deserts of China) (This figure was created with
ArcGIS).

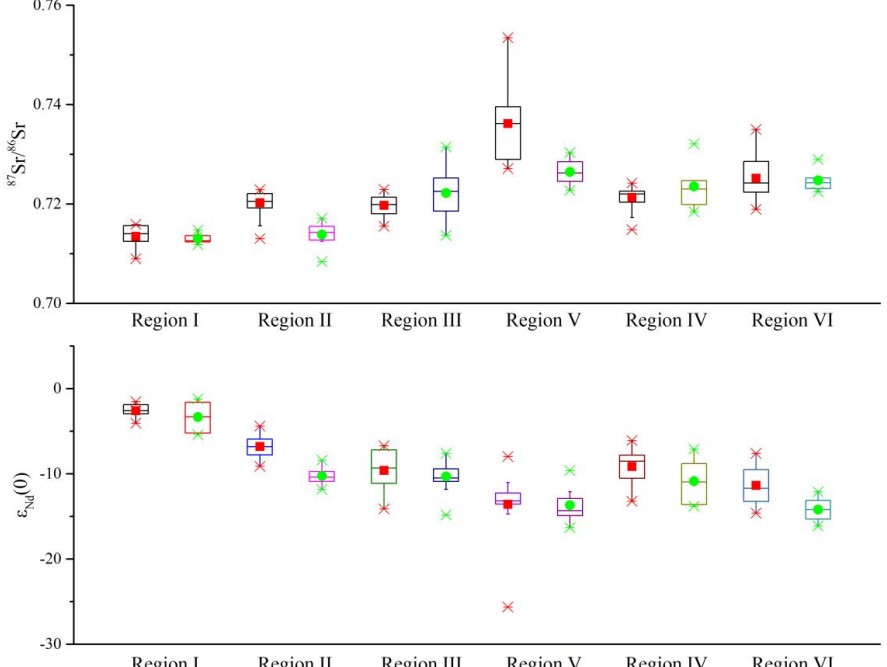


Fig. 3. Box plot for the Sr-Nd isotopic signatures of third pole PSAs and snow

samples. Samples are located in each PSA region based on the data from Table 2. The

mean Sr-Nd values are shown as red rectangles for sand or soil samples and green

solid cycles for snow samples.

**3.2. Sr-Nd data in natural dust or sediment from the Arctic**

Considerable Sr-Nd isotope data have been obtained from modern snow/ice

samples from the Arctic and surface and sea ice-transported sediments from the Arctic

Ocean, which covers the entire Arctic. Sand samples from PSAs (East Asian and

Saharan deserts) are also collected. Therefore, these data are useful for tracing the

terrigenous material transport for the Arctic. For user-friendly selection of the Sr-Nd

data according to the modern environment characteristics and the geographical

location, the 11 subregions were presented in the entire Arctic as follows (Fig. 4). The

patterns of the geographical zone have been similarly defined regarding seas and

drainage basins of the main river systems, as divided by Maccali et al. (2018).

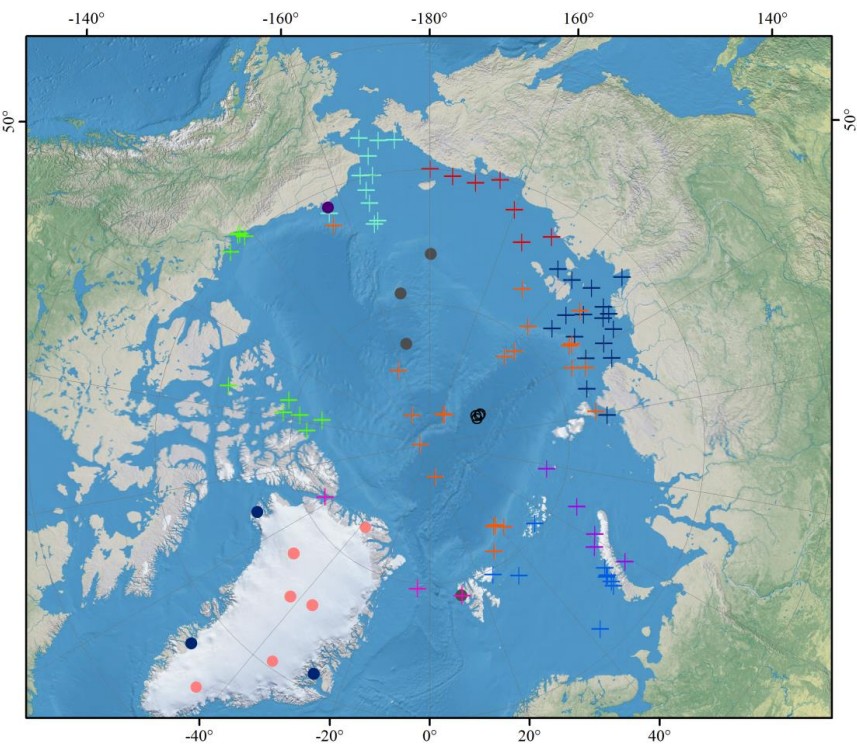


Fig. 4. Sampling distribution sites in the Arctic. The snow and cryoconite (sand or soil)
sampling sites are denoted with circles, and sediment samples are marked with
crosses (Table 3). 1. GrISS (Greenland ice sheet snow samples with orange solid
cycles); 2. GrIS-SSS (Greenland ice sheet sand or sediment samples with black solid
cycles); 3. SV (Svalbard sand or soil samples with magenta crosses); 4. AOS (Arctic
Ocean snow samples with grey solid and black cycles); 5. AOSS (Arctic Ocean sea
ice sediment samples with orange crosses); 6. BS (Barents Sea sediment samples with
blue crosses); 7. KS (Kara Sea sediment samples with violet crosses); 8. LS (Laptev
Sea sediment samples with black crosses); 9. ESS (East Siberian Sea sediment
samples with red crosses); 10. BCS (Bering-Chukchi Sea sediment samples with cyan



crosses); 11. MCAA (Mackenzie-Canadian Arctic Archipelago sediment samples with
green crosses) (This figure was created with ArcGIS).
**3.2.1 Sr-Nd data from snow/ice and sand samples of the Greenland ice sheet**

Sr-Nd data from different types of samples (snow/ice and sand samples) of

samples were collected from the GrIS. The snow samples were measured in the bulk
or >0.2 μm (0.45 μm) fraction, but it is noted that some shallow ice core samples from
the GrIS were filtered through two precleaned 0.2 μm and 30 μm filters (Lupker et al.,
2010). For some ice core samples, dust was subsequently extracted by evaporation
(Bory et al., 2003a). Sand samples were sieved to <71 μm in this study, and soil,
cryoconite, moraine, and englacial dust samples were used in bulk (Nagatsuka et al.,
2016). Sr-Nd data from the East Greenland Ice Core Project (EGRIP) and the North
GRIP (NGRIP) were measured via snowpits. Ice core samples were collected from
GRIP, GISP2, NEEM and RECAP. Sr-Nd data exhibit large differences between snow
and ice core samples (Fig. 5, Tables 1 and 3). In addition, the shallow ice core
samples were collected from Renland, Site A, Hans Tausen and Dye 3 (Du et al.,
2019b; Bory et al., 2003a). Cryoconite, moraine, and englacial dust samples were
collected from Kangerlussuaq, Thule, Scoresby Sund and Kong Christian X Land of
the GrIS (Nagatsuka et al., 2016; Simonsen et al., 2019).

The Sr-Nd data indicated that the dust sources were variable and showed

complicated dust sources in the same location for NGRIP snow (Bory et al., 2002;
Bory et al., 2003b). As much more Sr-Nd data from the sand, soil, cryoconite,
moraine, and englacial dust samples in the peripheral of the GrIS were measured,
$^{87}Sr/^{86}Sr$ values are the highest and the $^{143}Nd/^{144}Nd$ ratios are the least radiogenic in
these samples (Table 3). Compared with Sr-Nd data in NGRIP and EGRIP snowpits,
much larger variations were observed for $\varepsilon_{Nd}(0)$ in the EGRIP snowpit, and relatively



larger $^{87}$Sr/$^{86}$Sr values were observed in the NGRIP snowpit (Bory et al., 2002; Bory
et al., 2003b). Although the Sr-Nd isotopic ratios indicated Asian deserts might be the
main dust source for the GrIS. The ice-free region around the GrIS might be another
source for the interior GrIS. The Sr and Nd isotopic data in sediment samples
collected from the Scoresby Sund region by Simonsen et al. (2019) are as follows: the
$^{87}$Sr/$^{86}$Sr ratios range from 0.709689 to 0.736137, and the $\varepsilon_{Nd}(0)$ values range from
-15.7 to -10.1. Combining Sr-Nd values in snow (Renland, Site A, Hans Tausen and
Dye 3) and Dye 3 shallow ice core samples, the results showed that local dust sources
may contribute some of the    dust to the inland regions and that the Sahara is also the
most likely additional source (Lupker et al., 2010). Therefore, the local dust for the
free ice of the GrIS is another dust source, which may have been neglected in
previous studies.
To compare the data during the same period, therefore, Sr-Nd data from the deep
ice core were not included in Fig. 6. The mainstream view of the provenance of dust
in inland Greenland deep ice cores (GISP2 and GRIP) is that the dust is from the
eastern Asian deserts (the Gobi and Taklamakan Deserts) based on the best
geochemical matches during the last glacial period (Biscaye et al., 1997; Svensson et
al., 2000; Újvári et al., 2015). High-resolution Sr isotope data from the Greenland
NEEM ice core suggested that there was a significant Saharan dust influence in
Greenland during the last glacial period (Han et al., 2018). The Sr-Nd data (>5 μm) in
Holocene RECAP ice core samples are attributed to proximal dust sources; however,
the resolution of the data is approximately one thousand years (Simonsen et al., 2019).
In addition, the Sr-Nd data in Greenland deep ice core samples, which have low
resolutions and represent multiyear averages with no seasonal or interannual
variations (60 to 200 cm or 30-150 years), need to be considered (Biscaye et al., 1997;



Svensson et al., 2000).

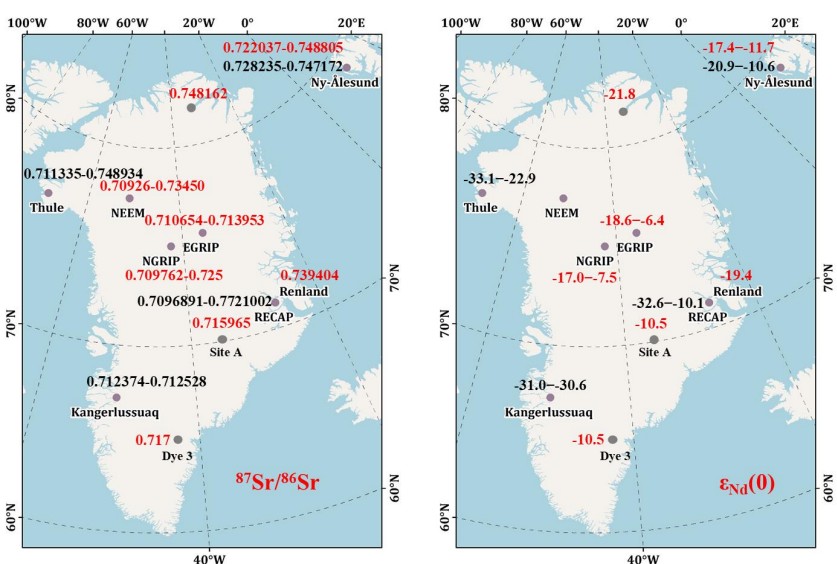


Fig. 5. $^{87}$Sr/$^{86}$Sr and $\varepsilon_{Nd}(0)$ values of insoluble dust in snow or ice core (red) and
sand/soil (black) samples from the Ny-Ålesund, Svalbard and GrIS (Bory et al., 2002;
2003a; Svensson et al., 2000; Biscaye et al., 1997; Lupker et al., 2010; Nagatsuka et
al., 2016; Han et al., 2018; Du et al., 2019b) (This figure was created with ArcGIS).

### 3.2.2 Sr-Nd data from snow and sediment samples in the Arctic Ocean

The $^{87}$Sr/$^{86}$Sr values are higher and $\varepsilon_{Nd}(0)$ values are lower in snow and sand

samples from Ny-Ålesund, Svalbard (SV), snow samples were measured in bulk and
sand samples were measured in the <71 µm fraction (Fig. 6, Du et al., 2019b). The
Sr-Nd data in snow samples from sea ice were measured in bulk, and four of these
samples were collected from near the North Pole in the western Arctic Ocean by
MOSAIC (October 2020) in this study (Figs. 4 and 6). However, the new $\varepsilon_{Nd}(0)$ data
have much more negative $\varepsilon_{Nd}(0)$ values (-20.8 to -19.6), which are very different from
previous results and cannot be explained by low latitude potential dust sources (Du et
al., 2019b). As shown in Fig. 6, the lowest ε$_{Nd}$ values were observed along the ice-free
periphery of the GrIS and SV; therefore, these ice-free regions are potential dust
sources for natural dust in the Arctic Ocean.

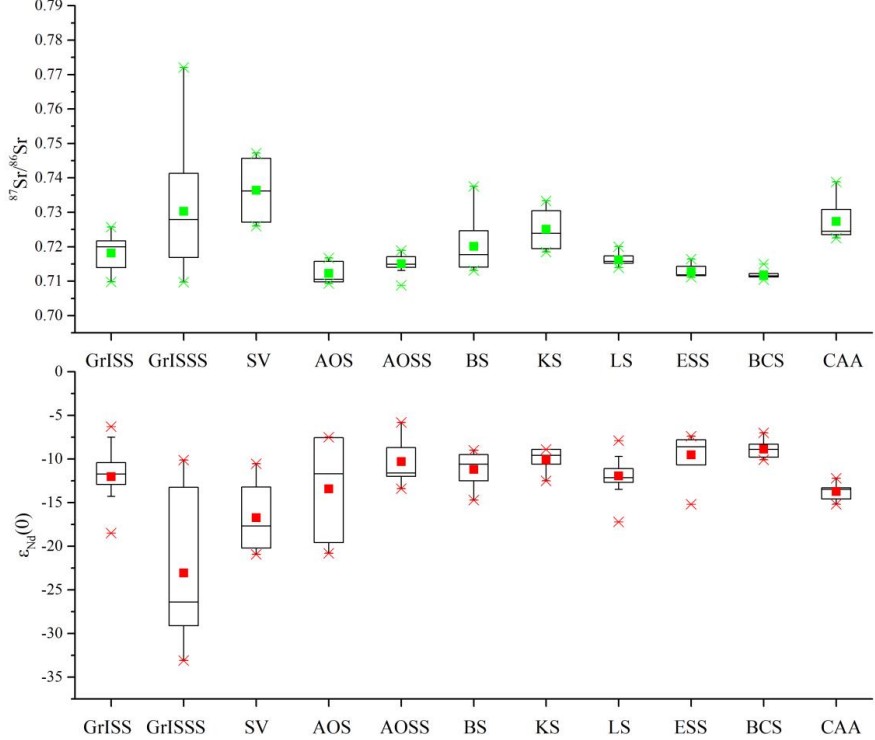


Fig. 6. Box plot for the Sr-Nd isotopic signatures of the Arctic, including the 11
subregion samples of snow, sand, soil, sediment from sea ice and sediment cores.

The Sr-Nd data from Arctic shelf surface sediment were based on the literature

(Fig. 6), and the Sr-Nd isotopic compositions in most samples were sieved at < 45 μm
for bulk. These data were chosen at water depths < 200 m and the surface or 0-10 cm
from the top to better represent the characteristics of coastal terrestrial sources
(Maccali et al., 2018). The Sr-Nd data in the BS, KS, LS, ESS, BCS and MCAA
surface sediment cores corresponded mainly to drainage basins and their adjacent seas
(Maccali et al., 2018). The sample spatial coverage in each region is variable, and Fig.



6 shows the relative homogeneity/heterogeneity for each region as well as isotopic
value overlaps. The box plot for these regions of the Arctic are further extended.
**3.3. Information on Sr-Nd data from the SH and Antarctic ice sheet**

By integrating the literature and adding data with new evidence, the pretreatment

method and characteristics of Sr-Nd are discussed in low-latitude SH. Dust
provenances of low-elevation areas on the periphery of the Antarctic ice sheet in the
modern and paleoenvironment were discussed. This study provides a comprehensive
overview of the state of knowledge of dust sources in different sectors of the Antarctic
ice sheet and PSAs in the SH under modern and ancient environments.
**3.3.1 Sr-Nd characteristics of SH potential dust sources**

There are three PSAs in the SH, including Australia, southern South America

(hereafter SSA) and Southern Africa (SA)). The Sr and Nd isotopic measurements
were taken exclusively on the <5 μm grain size fraction of Australian dust samples.
$^{87}Sr/^{86}Sr$ ratios ranging from 0.709 to 0.732 and $\varepsilon_{Nd}(0)$ values in Australian dust
samples between -15 and -3, some samples were well-dated samples deposited during
glacial periods (Revel-Rolland et al., 2006). Gaiero (2007) identified three main dust
sources in SSA with different grain sizes (bulk, 63 μm, 5 μm): the Puna
(22–26 °S)/Altiplano (15–22 °S) Plateau, with less radiogenic Nd (-9–-6 for $\varepsilon_{Nd}(0)$);
central-western Argentina (27–35 °S), with more radiogenic Nd (-6–-2 for $\varepsilon_{Nd}(0)$); and
Patagonia (39–52 °S) with more radiogenic Nd (-1–1 for $\varepsilon_{Nd}(0)$). The isotopic
compositions of aeolian dust from Argentina and Chile are confined to the ranges of
0.7045< $^{87}Sr/^{86}Sr$ <0.7130 and -5 <$\varepsilon_{Nd}(0)$ <3. The $^{87}Sr/^{86}Sr$ ratios in sand samples
from SA varied between 0.712348 and 0.74716, and the $\varepsilon_{Nd}(0)$ ratios varied between
-24.5 and -8.4 (Delmonte et al., 2003; Gili et al., 2021). In addition, the Sr and Nd
isotopic compositions in New Zealand sand samples were measured in the <5 μm





fraction according to Stoke's Law through humid sedimentation and separated from
PSA samples using a syringe; the values ranged from 0.70518 to 0.72324 for $^{87}Sr/^{86}Sr$
and -7.2 to -1.2 for $\varepsilon_{Nd}(0)$ (Delmonte, 2003). The fine fraction (<5 μm) of the Namibia
Sand Sea samples of SA was also separated following Stoke's Law by Gili et al.
(2021). The Sr-Nd data from PSAs were primarily measured by two references
(Delmonte et al., 2003; Gili et al., 2021). The measurement method is almost the same,
and these data can very clearly distinguish geographic subgroups for PSAs in SH.
**3.3.2 Sr-Nd data on the periphery and interior of the Antarctic ice sheet**

Sr-Nd data from Antarctica surface snow layers with different thicknesses

(6.5-10 cm) were measured with a three-metre long and one-metre wide snow pit
using a stainless steel saw (Bory et al., 2010). Sr-Nd data along the Zhongshan-Dome
A transect at 5–6 cm thick were measured using a Teflon shovel and the snow samples
were placed in 5 L Whirl-Pak bags (approximately 4–5 bags were collected at each
site) (Du et al., 2018). The $\varepsilon_{Nd}(0)$ values of marine sediment (near-core-top samples,
63 μm) from seven sectors of Antarctica are presented in Fig. 7 (Hemming et al.,
2007). Sr-Nd isotope data from coastal and low-elevation sites were also collected in
ice-free areas near the Filchner–Ronne Ice Shelf, Ross Ice Shelf and Amery Ice Shelf
(Fig. 7). The Sr and Nd isotopic compositions of four sand samples from southern
King George Island (South Shetland Islands) in West Antarctica, with less radiogenic
$^{87}Sr/^{86}Sr$ values ranging from ∼0.703907 to ∼0.704157 and $\varepsilon_{Nd}(0)$ values ranging
from 4.6 to 6.4, are relatively close to the $\varepsilon_{Nd}(0)$ values of marine sediment
(near-core-top samples) from the Antarctic Peninsula (ranging from -3 to 1)
(Hemming et al., 2007). The $^{87}Sr/^{86}Sr$ ratios ranged from 0.71135 to 0.72377, and the
$\varepsilon_{Nd}(0)$ composition ranged from -13.3 to -9.6 from ice-free areas of Inexpressible
Island in the Ross Sea, West Antarctica. The measurement method of sand samples



(<71 μm fraction) from southern King George Island and Inexpressible Island is the
same as that for the Zhongshan and Progress station sand samples from the Amery Ice
Shelf (Du et al., 2018). Based on the Sr-Nd data in our own and the literature (Table
4), four regions were divided in the SH:
Region A: McMurdo, King George Island and SSA (Patagonia), with the highest $\varepsilon_{Nd}(0)$
value of >-5.0;
Region B: Victoria Land, southern Australia, New South Wales, SA and SSA
(Puna–Altiplano area), with large variations in $^{87}Sr/^{86}Sr$ values and moderate $\varepsilon_{Nd}(0)$
values, and Taylor Glacier zero-age ice samples are very close to those of samples
from Victoria Land and the McMurdo dry valleys;
Region C: northern Australia, Victoria Land sources (including Inexpressible Island)
and SA, with high $^{87}Sr/^{86}Sr$ ratios and low $\varepsilon_{Nd}(0)$ values;
Region D: SA, northern Australia and the Amery Ice Shelf, with the lowest $\varepsilon_{Nd}(0)$
values of <-15.
Among these regions, although the data have significant differences, the Sr and
Nd compositions of some of the endmembers are similar, and care must be taken
when directly comparing these data to precisely explain the observed isotopic
compositions in ice core records. For example, there is overlap of the Sr and Nd
isotopic compositions of King George Island, SSA (Patagonia) and McMurdo dry
valleys. The new Sr and Nd data from Inexpressible Island also overlap with the other
endmembers (SA, New South Wale and Prydz Bay). Therefore, dust from low-latitude
regions (New South Wale and SA) cannot be excluded from East Antarctica (Du et al.,
2018; Gili et al., 2021). Another example is the characteristics of snow layers at the
Berkner Island ice sheet in western Antarctica. These data can be partly explained by
the surface sediment samples from the Weddell Sea sector, with $\varepsilon_{Nd}(0)$ values ranging
from -10 to -8 (Hemming et al., 2007). Therefore, the dataset from the SH and
Antarctic ice sheet demonstrates that multiple mixed sources can be inferred for
Antarctic surface snow samples. However, it should be noted that among the data
from the entire Antarctica ice sheet, Sr-Nd isotopic components were measured in
only 29 snow samples, and there is an urgent need to collect more data in the future.

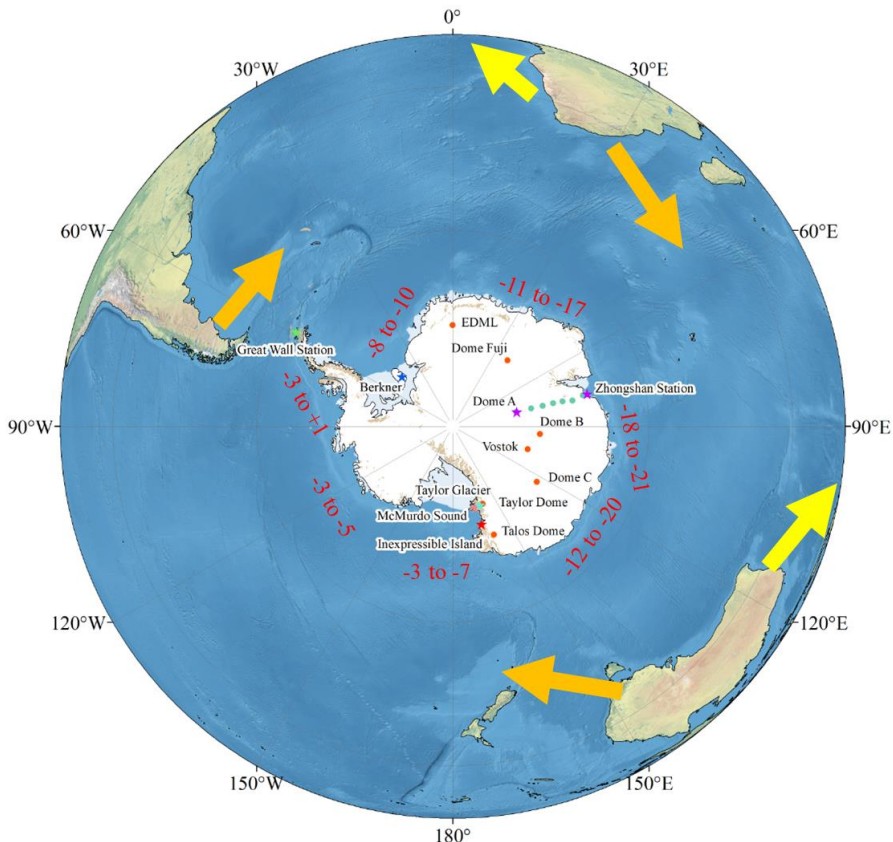


Fig. 7. Surface snow or snowpit samples are represented by purple solid circles, ice
core samples represent blue rectangles and samples represent red solid circles. Data
sources: Grousset et al., 1992; Basile et al., 1997; Delmonte et al., 2003, 2004, 2008,
2010, 2013 a and b; 2017, 2019; Gaiero, 2007; Revel-Rolland et al. (2006); Aarons et
al., 2016; 2017; Du et al., 2018. The sampling sites are noted with red circles (Table



S4). The dust transport paths are marked with yellow arrows. Each sector is
characterized by a range of $\varepsilon_{Nd}(0)$ values by S. R. Hemming et al. (2007) (This figure
was created with ArcGIS).

### 3.3.3 Sr and Nd data from Antarctic deep ice cores

Information on Sr-Nd data in Antarctic ice cores during the Holocene and

glacial-interglacial times is presented by integrating literature (Table 4). To obtain
Sr-Nd data, mineral dust from EPICA-Dome C and Vostok ice core was extracted by
filtration using 0.4 μm membranes. Each filter was put into a precleaned Corning tube
filled with ~10 mL Milli-Q water, and microparticles were removed from the filter
through sonication. The liquid was then evaporated in a clean hood dedicated to
chemical preparation of samples for Sr-Nd analyses (Delmonte et al., 2008). To obtain
enough dust particles, the different age interval samples were merged. For example,
each sample represents approximately 40-160 years for the Vostok ice core, which is a
few thousand years to obtain a single large-volume sample (Delmonte et al., 2008).
Alternatively, several ice core sections from different depths were integrated to obtain
a few large samples for the Sr and Nd isotopic analyses of the Talos Dome ice core
(Delmonte et al., 2010b). A relatively high resolution (spanning between ∼ 3 and ∼
30 yrs.) was used in the Taylor Dome ice core Sr and Nd were measured using a
TIMS equipped with $10^{11}$ Ohm resistors for $^{87}Sr/^{86}Sr$ ratios and $10^{13}$ Ohm resistors for
$^{143}Nd/^{144}Nd$ ratios (Aarons et al., 2016). Aarons et al. (2017) measured Sr-Nd data in a
horizontal ice core of the ablation area from the Taylor Glacier in East Antarctica,
which was decontaminated and processed for    each ~20 kg ice core sample.

Ice cores are very difficult to obtain and measurement methods limit continuous

data. Sr-Nd data in the Antarctica deep ice core mainly focus on the coastal and inland
areas of the EAIS. As previously mentioned, the dust source is similar to that of the

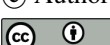



modern samples in the Dome C and Vostok ice cores during the Holocene and
interglacial periods, which can be explained by an SSA provenance; an additional
hypothesis explains the isotopic signature of Holocene dust in central East Antarctica
(Delmonte et al., 2008, Delmonte et al., 2019). The Sr-Nd data in the Talos Dome,
Taylor Dome and Taylor Glacier ice cores during the Holocene point towards a local
dust provenance (Delmonte et al., 2019; Aarons et al., 2016, 2017). Therefore, the Sr
and Nd data from East Antarctica ice cores during the Holocene and interglacial
periods indicate a well-mixed atmospheric background involving a mixture of two or
more sources in the SH (Fig. 8). The newest study demonstrated that SA emerges as
the second most important dust source to East Antarctica during interglacial periods
(Gili et al., 2021).
However, the glacial stage (stage 4: ~ 60 ka and stage 6: ~ 160 ka) samples in the
Vostok ice core span a very narrow range of Sr compositions ($0.708219 < {}^{87}Sr/{}^{86}Sr <$
$0.708452$) and Nd compositions ($1.1 < \varepsilon_{Nd}(0) < 5.0$), which can also be explained by
the new Sr-Nd data in sand samples (<71 μm) from southern King George Island
(${}^{87}Sr/{}^{86}Sr$ values ranging from ~0.703907 to ~0.704157 and $\varepsilon_{Nd}(0)$ values ranging
from 4.6 to 6.4). The ${}^{87}Sr/{}^{86}Sr$ and $\varepsilon_{Nd}(0)$ isotopic compositions of dust in the Taylor
Glacier ice core samples during the last glacial period indicated that dust may
originate from SSA and from potential local source areas in the Ross Sea Sector
(Delmonte et al., 2010; Aarons et al., 2016; Aarons et al., 2017). Therefore, these data
suggest that the glacial-period dust in East Antarctic ice cores also contributes from
local contributions (Fig. 8). However, almost no Sr-Nd data were from the West
Antarctic deep ice cores, which limits to understand the dust transport in the spatial
and temporal distribution of the entire Antarctic ice sheet.

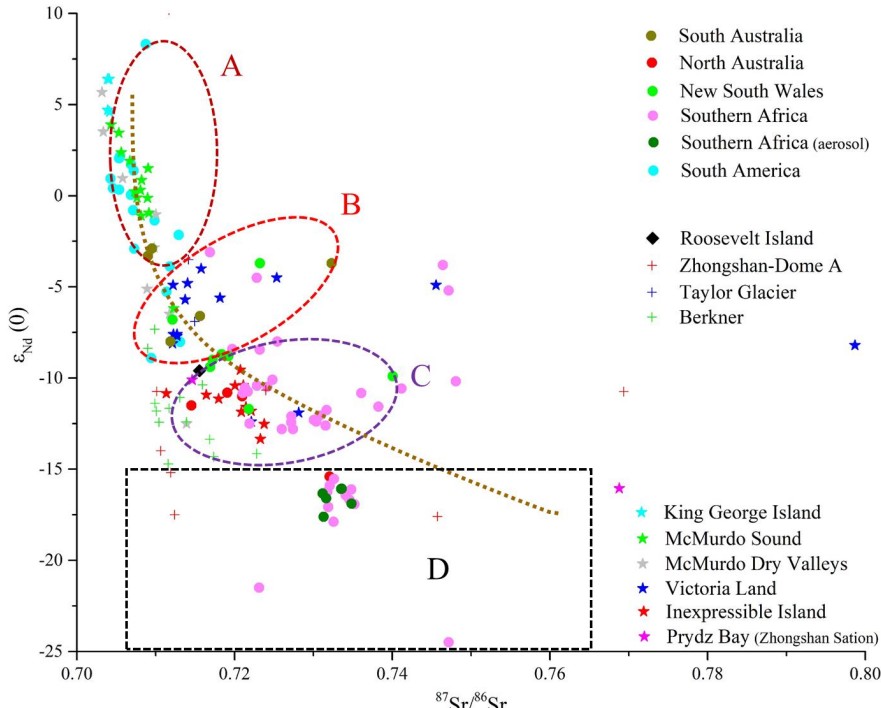


Fig. 8. Sr isotopic composition versus Nd isotopic composition of modern dust
deposited in potential dust source areas of Australia (including New South Wales,
Revel-Rolland et al., 2006; Grousset et al., 1992), southern Africa (Grousset et al.,
1992; Delmonte 2004a; Gili et al., 2021) and South America (Delmonte 2004a) (solid
circles with different colours) at the peripheral Antarctic ice sheet of ice-free areas
(King George Island and Inexpressible Island, this study; McMurdo Sound, Winton et
al., 2014, 2016b; McMurdo Dry Valleys, Gemmiti (2000); Victoria Land, Delmonte et
al., 2013; Prydz Bay, Du et al., 2018; stars with different colours), and data obtained
on surface snow samples of Antarctic ice sheets with differently coloured crosses
(data of Roosevelt Island fromWinton et al., 2016a; data of Zhongshan-Dome A from
Du et al., 2018; data of Taylor Glacier from Aarons et al., 2017; data of Berkner from
Bory et al., 2010).
**4. Data availability**



All datasets and the associated metadata table presented in this study are available
through a Big Earth Data Platform for Three Poles. The dataset can be downloaded
from https://doi.org/10.11888/Cryos.tpdc.272100) (Du et al., 2022). In this repository,
the entire datasets are provided in Excel spreadsheet format together with metadata
files.
**5. Conclusions**
The maintenance integrated Sr-Nd dataset was presented from the remote three
poles, and these data are not easily collected because of the extremely cold
environment. The important conclusion to be drawn from this study is that
understanding the dust transport paths of the three poles in a warming environment
exposes large source areas of dust. The dataset is complicated and includes snow,
sand, soil, loess, deposits, sediment and other types. These integrated data can provide
a new perspective into present and paleodust sources from the three poles, more
importantly, which clearly emphasizes the following points for potential users of the
datasets provided with this paper:
1. This Sr-Nd dataset enables us to map the standardized locations in the remote
three poles, while the use of sorting criteria related to the sample location, type or
resolution permits us to trace the dust source and sink based on their isotopic
signatures.
2. For the third pole, each subregion of Sr-Nd data was provided, and the data will
be useful for tracing the local or long-distance transported dust of the source and
sink. The integration of these data between sand (soil) and snow samples for six
subregions allowed us to clearly understand the Sr-Nd data characteristics in the
third pole.
3. There are 11 subregions for the entire Arctic, and Sr-Nd data can provide the user



with sink information on dust in the Arctic environment, which would be useful
for tracing dust sources for the Arctic.
4.  The new data from Arctic and Antarctica samples emphasized the ice-free regions
on the periphery of the ice sheets, which may be important local dust sources. In
particular, Sr-Nd data overlap with the low-latitude region of PSAs. However, the
paucity of data in Antarctica is serious and future studies should concentrate on
this aspect.

**Author contributions**. CX, ZD, and SA designed the study, ZD, JY, CX and SA
wrote the manuscript. ZD, LW, NW, SW, YL collected the samples in field and
produced data. ZD, NW, LW, SW, YL, ZW, XM performed analysis. All authors
contributed to the final form of the manuscript.
**Competing interests.** The authors declare that they have no conflict of interest.
This work was supported by the Strategic Priority Research Program of the Chinese
Academy of Sciences (XAD19070103), the National Natural Science Foundation of
China (Grant Nos. 42071086 and 41971088), the Youth Innovation Promotion
Association, CAS (2020419) and the State Key Laboratory of Cryospheric Science
(SKLCS-ZZ-2021).

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






Table 1. Data distribution locations and sample types for $^{87}Sr/^{86}Sr$ and $\varepsilon_{Nd}(0)$ from 42 references.

| Location | Sample type | Data points | $^{87}Sr/^{86}Sr$ | $\varepsilon_{Nd}(0)$ | References |
|---|---|---|---|---|---|
| Third Pole | | 274 | | | |
| Altai Mountains | Snow/Sand | 12 | Yes | Yes | Chen et al., 2007; Xu et al., 2012; Du etal., 2019a |
| Tienshan and Kunnun | | | | | Chen et al., 2007; Nagatsuka et al., 2010; Xu et al., |
| Mountains (including | Snow/Ice/sand | 45 | Yes | Yes | 2012; Du et al., 2015; Du et al., 2019a; Wei et al., |
| Central Tibet Plateau | Snow/Sand | 39 | Yes | Yes | Xu et al., 2012; Du et al., 2019a; Wei et al., 2019 |
| Himalaya Mountains | Snow/Sand | 34 | Yes | Yes | Xu et al., 2012; Du et al., 2019a; Wei et al., 2019 |
| Qiliuan Mountains | Snow/Ice/Sand | 50 | Yes | Yes | Wu et al., 2010; Xu et al., 2012; Dong et al., 2018; Du |
| Heduan Mountains | Snow/Sand | 24 | Yes | Yes | Xu et al., 2012; Dong et al., 2018; Wei et al., 2021 |
| Other deserts and loess in | | | | | |
| China | Sand/Loess | 70 | Yes | Yes | Chen et al., 2007; Du et al., 2019a |
| Arctic | | 205 | | | |
| | | | | | Biscaye et al., 1997; Svensson et al., 2000; Bory et al., |
| | | | | | 2002; Bory et al., 2003a; Lupker et al., 2010; |
| | | | | | Nagatsuka et al., 2016; Han et al., 2018; Simonsen et |
| Greenland | Snow/Ice/Cryoconite/Sediment | 186 | Yes | Yes | al., 2019 |
| Svalbard | Snow/Sand/Loess | 7 | Yes | Yes | Du et al., 2019b |
| Arctic Ocean | Snow/Sediment | 104 | Yes | Yes | Maccali et al., 2018; Du et al., 2019b; this study |
| Alaska | Snow/Sand/Loess | 5 | Yes | Yes | Du et al., 2019b |
| Antarctica | | 354 | | | |
| | | | | | Paolo et al., 1982; Grousset et al., 1992; Basile et al. |
| | | | | | 1997; Gemmiti, 2000; Delmonte, 2003; Delmonte et |
| | | | | | al., 2004 a, b; Delmonte et al., 2007; Delmonte et al., |
| | | | | | 2008; Delmonte et al., 2010a; Delmonte et al., 2013; |
| | | | | | Aarons et al., 2016; Winton et al., 2016 a, b; Delmonte |
| East Antarctica | Snow/Sand | 215 | Yes | Yes | et al., 2017; Aarons et al., 2017; Du et al., 2018 |
| | Snow/Ice/Sand/Aeolian | | | | Bory et al., 2010; Delmonte et al., 2010b; Winton et |
| West Antarctica | deposits/Rock | 27 | Yes | Yes | al., 2016a; this study |
| Southern American | Loess/Soil/Sediment/Aeolian | 57 | Yes | Yes | Grousset, et al., 1992; Basile et al., 1997; Delmonte, |
| | | | | | Grousset, et al., 1992; Delmonte, 2003; Gili.,et al., |
| Southern Africa | Aeolian dust/Loess | 24 | Yes | Yes | 2021 |
| Australia | Sand/Loess/Sediment | 15 | Yes | Yes | Grousset, et al., 1992; Revel-Rolland et al., 2007 |



 

Table 2. Snow, sand and soil samples were located in the third pole glaciers and PSAs of dust generation. Headers from left to right: Label: the number of glaciers; Subregions; Glacier name; Site name: name of the sampling site where the samples were taken; Longitude and Latitude; sampling location; Sample type: Snow, sand or soil; Elevation: m a.s.l; Isotopic ratios of Sr and $\varepsilon_{Nd}(0)$; Ref.: reference publications. The different colours represent different subregions.

| Label | Sub-regions | Glacier name | Site name | Latitude (°N) | Longitude (°E) | Mountains | Sample type | Elevation (m a.s.l) | $^{87}Sr/^{86}Sr$ | esa(0) | Ref |
|---|---|---|---|---|---|---|---|---|---|---|---|
| 1 | Region I | Muzidao | MSD | 47°06'N | 85°33'E | Altai mountains | Snow | 3605 | 0.713185~0.713871 | -6.55~-4.80 | Xu et al., 2012 |
| 2 | | Muztagata | MS | 38°17'N | 75°06'E | East Pamirs | Snow | 6565 | 0.717187~0.717415 | -10.3~-8.4 | Xu et al., 2012 |
| 3 | Region II | Tianshan No. 1 | TS/UM | 43°07'N | 86°49'E | Tien Shan | Snow/Surface dust | 4063 | 0.719404~0.721728 | -10.9~-6.9 | Nagatsuka et al., 2010; Xu et al., 2012 |
| 4 | | Miaoergou | MG | 43°03'19"N | 94°19'21"E | Tien Shan | Ice/Snowpack/Cryoconite | 3100~4512 | 0.710284~0.720825 | -11.6~-7.3 | Du et al., 2015; Wei et al., 2019 |
| 5 | | Yuzhufeng Glacier | YG-1 | 35°39'37" N | 94°14'20"E | Kulun Mountains | Snow | 4300~4720 | 0.714821~0.716757 | -16.6~-11.8 | Wei et al., 2019 |
| 6 | | Zangsepangri | ZSGR | 34°16'N | 85°51'E | Qiangtang plateau | Snow | 6228 | 0.717352~0.718328 | -12.9~-9.2 | Xu et al., 2012 |
| 7 | Region III | Guoqu | GL | 33°35'N | 91°52'E | Tanggula mountains | Snow | 5760 | 0.717566~0.721786 | -10.2~-9.5 | Xu et al., 2012 |
| 8 | | Dongkemadi | DKMD | 33°06'N | 92°06'E | Tanggula mountains | Snow | 5700 | 0.713192 | -10.51 | Xu et al., 2012 |
| 9 | | Zadang | ZD | 30°28'N | 90°39'E | Nyainqentanglha | Snow | 5758 | 0.718285~0.721305 | -12.9~-11.1 | Xu et al., 2012 |
| 10 | | Jiemayangrong | JMYZ | 30°13'N | 82°10'E | Himalaya | Snow | 5558 | 0.72671~0.740694 | -14.3~-10.5 | Xu et al., 2012 |
| 11 | Region IV | Yala | Yala | 28°34'E | 85°37'N | Himalaya | Snow | 5190 | 0.740112 | -15.68 | Xu et al., 2012 |
| 12 | | East Rongbuk | QM | 28°06'N | 86°58'E | Himalaya | Snow | 6525 | 0.728057~0.757407 | -28.1~-14.7 | Xu et al., 2012 |
| 13 | | Laohugou Glacier No.12 | LHG | 39°26'N | 96°32'E | Qilian Mountains | Snow | 4288~5026 | 0.720448~0.723303 | -15.7~-9.5 | Xu et al., 2012; Wei et al., 2019 |
| 14 | | Dunde ice cap | DD | 38°06'N | 96°24'E | Qilian Mountains | Ice | 5325 | 0.715220~0.721074 | -11.1~-9.9 | Wu et al., 2010 |
| 15 | | Qiyi Glacier 1 | QG | 39°54'1.5"N | 97°45'20"E | Qilian mountains | Snow | 4500~4750 | 0.712349~0.722751 | -13.3~-8.6 | Dong et al., 2018 |
| 16 | Region V | Shiyi Glacier | SD | 38°52'45"N | 99°52'40"E | Qilian mountains | Snow | 3928~4152 | 0.721032~0.721711 | -14.0~-13.8 | Wei et al., 2019 |
| 17 | | Dahanshan | DS | 37°21'43"N | 101°24'12" E | Qilian mountains | Snow | 3593~3625 | 0.723105~0.725015 | -12.1~-12.0 | Wei et al., 2019 |
| 18 | | Lenglongling Glacier | LG | 37°31'N | 101°54'E | Qilian mountains | Snow | 3558~3992 | 0.719084~0.728414 | -10.9~-7.0 | Dong et al., 2018 |
| 19 | | Dagu Glacier | DG | 32°07' N | 102°26' E | Hengduan mountains | snow | 3520~3701 | 0.719216~0.721102 | -16.9~-12.3 | Dong et al., 2018 |
| 20 | | Haihorgou Glacier | HG | 29°20' N | 101°54' E | Hengduan mountains | snow | 3010~3850 | 0.722805~0.728826 | -17.1~-12.0 | Dong et al., 2018 |
| 21 | Region VI | Demula Glacier | DML | 29°22' N | 97°00' E | Hengduan mountains | Snow | 5404 | 0.729095~0.735863 | -17.1~-14.2 | Xu et al., 2012 |
| 22 | | Baishui Glacier No.1 | YL | 27°06' N | 100°12' E | Hengduan mountains | Snow | 4338~4747 | 0.717145~0.719881 | -13.8~-11.4 | Xu et al., 2012; Dong et al., 2018 |





Table 3. Snow, cryoconite, sand, soil and sediment samples located in the Arctic. Headers from left to
right: Label; Subregions; name of the sampling site where the samples were taken; Sample type: Snow,
Cryoconite, sand and soil; Elevation: m a.s.l; Ref.: reference publications.

| Label | Subregion | Location | Sample type | Elevation (m a.s.l) | Time interval | Size fraction | Ref |
|---|---|---|---|---|---|---|---|
| 1 | GrISS | East Greenland | snowpit | 2702 | 2017/2018 | >0.2 µm | This study |
| 1 | GrISS | North Greenland | snowpit | 2959 | early-1995 | <45 or 38 µm | Bory et al., 2002 |
| 2 | GrISSS | Central East Greenland | Rock, Powder, Sediment | NO | No | bulk | Simonsen et al., 2019 |
| 2 | GrISSS | West Greenland | Cryoconite, Moraine, Englacial dust, Sand | 247 | No | bulk | Nagatsuka et al., 2016; This study |
| 3 | SV | Arctic Ocean | Sediment | 0 | No | <100 µm | Tütken et al., 2002; Maccali et al., 2018 |
| 3 | SV | Ny-Ålesund | Snow, Sand, Soil | 0-500 | 2016 | Bulk | Du et al., 2019b |
| 4 | AOS | Arctic Ocean | Snow | 0 | 2016 | Bulk | Du et al., 2019b; This study |
| 5 | AOSS | Arctic Ocean | Sea-ice sediment | 0 | No | bulk | Tütken et al., 2002; Maccali et al., 2018 |
| 6 | BS | Arctic Ocean | Sediment | 0 | No | <100 µm | Maccali et al., 2018 |
| 7 | KS | Arctic Ocean | Sediment | 0 | No | <100 µm | Tütken et al., 2002; Maccali et al., 2018 |
| 8 | LS | Arctic Ocean | Sediment | 0 | No | <100 µm; Bulk | Eisenhauer et al., 1999; Maccali et al., 2018 |
| 9 | ESS | Arctic Ocean | Sediment | 0 | No | <100 µm | Maccali et al., 2018 |
| 10 | BCS | Arctic Ocean | Sediment | 0 | No | <100 µm; Detrital | Asahara et al., 2012; Maccali et al., 2018 |
| 11 | CAA | Arctic Ocean | Sediment | 0 | No | <100 µm | Maccali et al., 2018 |









Table 4. Samples information in the SH and Antarctica. Headers from left to right name of the sampling
site where the samples were taken; sample numbers; time interval (age); Sample type; Size fraction;
Ref.: reference publications.

| Site name | Numbers | Time interval (age) | Sample type | Size fraction | Ref |
|---|---|---|---|---|---|
| Zhongshan Station-Dome A, East Antarctica | 11 | 2016/2017 | Snow | >0.2 µm | Du et al., 2018 |
| Roosevelt Island, West Antarctica | 2 | 2011/2012 | Snow | >0.2 µm | Winton et al., 2016a |
| Surface snow (top ~3 cm), West Antarctica | 14 | 2002/2003 | Snow | >0.4 µm | Bory et al., 2010 |
| ITASE, East Antarctica | 3 | 1420–1800 A.D. | Ice | Bulk | Delmonte et al., 2013 |
| Talos Dome, Dome C, Vostok, East Antarctica | 6 | Holocene | Ice | >0.4 µm | Basile (1997); Delmonte et al., 2010a |
| Komosmolskaia ice core, East Antarctica | 2 | Deglaciation | Ice | Bulk | Delmonte et al., 2004b |
| Talos Dome, Dome B, Dome C, Old Dome C, Vostok, East Antarctica | 56 | MIS 2, 3, 4, 5, 5.5, 6, 8, 10 and older | Ice | Bulk | Basile et al., 199; Grousset et al., 1992; Delmonte, 2003; Delmonte et al., 2004a; Delmonte et al., 2004b; Delmonte et al., 2008; Delmonte et al., 2010a; Delmonte et al., 2010b; Delmonte et al., 2017 |
| Taylor Glacier, East Antarctica | 39 | 0-45711 | Ice | >0.2 µm | Aarons et al., 2017 |
| Taylor Dome, East Antarctica | 34 | 1100-31400 | Ice | >0.2 µm and <30 µm | Aarons et al., 2016 |
| Bolivia, Illimani, Southern America | 7 | 1950-1930 A.D. | Ice | <8 µm | Delmonte et al., 2010b |
| Zhongshan and Progress stations, East Antarctica | 2 | n.d. | Sand | <71 µm | Du et al., 2018 |
| Inexpressible Island, East Antarctica | 11 | n.d. | Sand | <71 µm | This study |
| King George Island, West Antarctica | 4 | n.d. | Sand | <71 µm | This study |
| McMurdo Sound, East Antarctica | 11 | n.d. | Sediment trap | Bulk | Winton et al., 2016b |
| Ross sea, East Antarctica | 2 | n.d. | Sediment trap | Bulk | Winton et al., 2016b |
| Enderby Land, East Antarctica | 5 | n.d. | Rocks | Bulk | Paolo et al., 1982 |
| Mimy (continental shelf sediments), East Antarctica | 1 | n.d. | Sediment | Bulk | Basile et al., 1997 |
| Terre Adelie (continental shelf sediments), East Antarctica | 3 | n.d. | Sediment | Bulk, <5 µm | Grousset, et al., 1992 |
| Bunger Hills (moraines), East Antarctica | 2 | n.d. | Moraines | Bulk | Basile et al., 1997 |
| Dry valleys, East Antarctica | 1 | n.d. | Sand | <30 µm | Basile et al., 1997 |



| Victoria Land | 14 | n.d. | Regolith (granite) | <5 μm | Delmonte et al., 2013 |
|---|---|---|---|---|---|
| McMurdo Dry Valleys and | | | | | |
| Northern Victoria Land, | | | | | |
| East Antarctica | 11 | n.d. | Aeolian deposits | < 5 μm | Gemmiti, 2000, Delmonte, 2003 |
| | | | Loess, Soil, Sediment, | Bulk, <5 | |
| | | | Eolian dust, Volcanic | μm,  < 10 | |
| South America[*] | 57 | n.d. | materials, Aeolian deposits | μm, < 63 μm | Grousset, et al., 1992, Basile et al., 1997, Delmonte, 2003, Gaiero et al., 2007 |
| | | | Suspended dust, Loess, Sand | | |
| | | | dune, Lacustrine sediment, | | |
| | | | Sand dune, Loess-like | | |
| Australia[*] | 24 | n.d. | deposit, Marine sediment | < 5 μm | Grousset, et al., 1992, Revel-Rolland et al., 2007 |
| | | | Dust, Loess, Aerosol, | <5 μm, Fine, | |
| Africa | 53 | n.d. | Sediment; Aeolian deposits | Bulk | Grousset, et al., 1992, Delmonte, 2003, Gili et al., 2021 |
| | | | | Bulk or < 5 | |
| New Zealand | 16 | n.d. | Loesses, Aeolian deposits | μm | Basile et al., 1997;Gemmiti, 2000; Delmonte, 2003 |

* means few samples have the dating ages.