# Peer review of "A database of radiogenic Sr-Nd isotopes at the “three poles”"

_Earth System Science Data, 2022_

## Referee Comment (RC2)

Review for Du et al. ESSD by Cécile Blanchet

Potsdam, April 19th 2022

The authors present a data compilation for neodymium and strontium isotopes to trace dust deposition in the "three poles". In order to do that, the authors have assembled data from the literature as well as new measurements. I would recommend to reorganize the manuscript and better define the objectives of the study before this can be published. I have several comments, which I hope will help to improve the manuscript.

Main comments:
1) I am not entirely sure what the **main goal of the paper** is: is it to determine where dust deposited on the three poles comes from? Is it to determine how the three poles act as sources and/or sinks through time? Is it to determine the role of local dust sources to the sinks on the three poles? On line 106, it is not clear which are the "questions" the authors are aiming to answer. Clarifying the aim will help to understand the rationale of compiling data from soil surfaces, glacier, snow, and marine sediment cores, but not using more global databases that would allow to determine the contribution from other potential source areas.

2) If the aim of the paper is to describe the dataset, the authors need to **spend more time explaining their strategy for data collection**. Literature search? Use of previous compilations? Number of data from the literature versus samples measured and published in this paper. Needs to be clarified, esp. in lines 175-190, where it is quite confusing (although it is clearer in the abstract). Please separate in the methods the literature search from the sample collection and measurements.

3) Data file (excel spreadsheet): I struggle with the way the data are presented: I would recommend to use one template for all the data. In the attached spreadsheet, there are four different sheets with each a different template. **For the sake of being able to compare the data with each other, please follow a strict (and similar) order and include "NA"s when an information is missing**. If you want your data to be used by others, why not using the template that we published last year (version 3.0 of my database: https://dataservices.gfz-potsdam.de/panmetaworks/showshort.php?id=7124101c-d2a2-11eb-9603-497c92695674), with perhaps modifying the "water depth (m)" column into an "elevation (m)" column (with changing signs whether it is altitude (+, masl) or bathymetry (-, mbsl)? Also please **provide the locations in decimal degrees** as it allows most programs to plot the data (it is more difficult with locations in minute/second). Also please provide the metadata in the downloadable spreadsheet.

4) Data visualisation: I think that one of the powerful possibilities of compiling data and attributing precise locations is to explore the spatial variability of certain variables by plotting **isoscapes** (instead of plotting numbers on locations like in Fig. 5). In your case, it would be very interesting to compare the changes in Nd and Sr isotopes spatially in the three poles for the source material (soils) and for the sinks (snow, sediments). In general, the authors need to **better separate sources and sinks** throughout the manuscript.

5) Box and whiskers plots (Fig. 3 and 6): please indicate the number of samples used for each category. Please note that these analyses are best suited for >5 samples.

6) In general, the isotopic signature needs to be discussed in terms of lithologicak context, which is seldom mentioned in the manuscript. Perhaps use geological maps to contextualise the varying signatures observed?

7) There are way too many abbreviations and I must say, I got lost. Please use abbreviations only when necessary but otherwise use the full names.

Minor comments:
L. 29-30: "recognized and introduced": what do you mean here?

L. 115: what is "data augmentation"?

L. 122-123: it might be useful to clarify the expression "three poles" for the readers who might not be familiar with it. I knew the expression "third pole" but not "three poles". In lines 124-125, you cite Australia, Southern south America, Southern Africa and New Zealand: are these regions part of the three poles? I thought the third pole refers to the Himalayas? Please clarify.

L. 126: I don't think that the abbreviation TP has been introduced before.

L. 131: Please explain what a cryoconite sample is.

l. 182: Very pleased that you used the scheme I developed: it will be very useful to compare and compile datasets!

L. 218-219: I do not understand this sentence. The "acid leaching method" used is not given in the spreadsheets.

l. 220: "This feature validates" is a strange formulation, please clarify.

L. 221-222: how did you determine PSAs exactly? Based on which criteria?

L. 224-253: are these six regions the PSAs? How were they determined?

L. 301-306: belong to the methods. This happens repeatedly throughout the manuscript. Please all technical considerations should be put in the methods. However, the authors are often describing how other teams have sampled, which seems not really necessary.

It might be advisable to the authors to edit the paper: there are often redundancies or unclear sentences, which hinder a smooth reading and a comprehension of the paper.

L. 374-376: I do not understand this sentence. I cannot see any homogeneity/heterogeneity from Fig. 6. Please clarify or modify Fig. 6.

L. 377. Antarctica: there are a lot of new data for Antarctica assembled in the paper of Robinson et al. (2021) in Chemical Geology (Open Access, https://doi.org/10.1016/j.chemgeo.2021.120119). The accompanying database version is V 3.0: https://dataservices.gfz-potsdam.de/panmetaworks/showshort.php?id=7124101c-d2a2-11eb-9603-497c92695674)

L. 429-438: the separation in these different areas seems quite arbitrary. What is it based on? The authors mix sources and sink and it is not very helpful to understand the processes driving dust transportation I think.

L530-556. Conclusions

L. 540-543: I find this claim not really supported by the manuscript. The criteria were not used to determine which dust source was contributing to which sink. This needs to be better demonstrated, e.g., by separating sources and sink in the database or in the figures.

L. 554. A PSA should be defined based on present-day knowledge of dust generation, e.g., based on remote sensing or geomorphological evidence. At minima it needs to be defined by the dust produced: grain-size distribution, mineralogy, isotope signature as well as by its geographical location.

Figures:

I would encourage the authors to revise their figures to make them more informative.

Fig. 2: The numbers on the map (glaciers and deserts) need to be related to the names in a legend or in the caption.

Fig. 3: see my previous comment on box and whiskers plots.

Fig. 4: There needs to be a legend on this map to help the reader to identify the different sample sets.

Fig. 5: This figure would be much better is plotted as an isoscape (using interpolation between known values), and with maps for sources and maps for sinks. Actually, I think that Fig. 4 and 5 could be combined in isoscapes.

Fig. 6. See Fig.3. I struggle with the abbreviations: please put the full names.

Fig. 7: same as comment for Fig. 5.

Fig. 8: What are the A, B, C and D areas? Perhaps choose colour codes to distinguish the main areas?

Data: See main comment 3)

---

## Author Comment (AC1)

Answer to reviewer comment 1

Review of Du et al. A database of radiogenic Sr-Nd isotopes at the "three poles"

General comments

Du et al. present a dataset regarding the Sr and Nd isotope of various surface sediments to trace the source-to-sink process of dust in the three poles of the Earth. Such a data compilation is timely because there are emerging Sr-Nd isotopic data in past years. I list some concerns for improving this paper.

Reply: We appreciate that the reviewer finds our work interesting and acknowledge the suggestions for improvement. We have regrouped some of the questions and added much more data to improve the text flow.

First, a major pitfall of this paper is that many previous studies regarding Sr-Nd isotopes of surface sediment are absent in this study. I am only familiar with those studies on Asian dust sources, and the suggestion can be found in specific comments below. The authors should check other regions, e.g., the Arctic and Antarctic.

Reply: Thanks, we added the surface sediment data from the Third Pole, and Sr-Nd data from the Arctic and Antarctica were also collected in revised dataset (See in the dataset).

Second, the present compilation of Asian dust sources only covers the Tibetan Plateau and Jungar Basin. The data in other regions of central and East Asia should not be omitted in this study because those data in North China and Mongolia could provide dust to the North Pacific and even Greenland (dust in the Arctic).

Reply: Thanks, in original version, we just considered the high altitude regions. We added Sr-Nd data based on the area of Third Pole. And Sr-Nd data from the Pan-Third Pole were also added in revised manuscript.

Third, because the Sr-Nd isotopic fingerprint in Asian dust sources was studied extensively, the paper should not only focus on the bulk sediment Sr-Nd isotopic but also leave a position for the different grain sizes (e.g., for Nd and Sr isotopes) and different mineral phases (for Sr isotopes), those distinct grain size and phases will correspond to different dust dynamics.

Reply: Thanks for your suggestions, we add Sr-Nd data, which referred the different

grain sizes and mineral phases Sr-Nd data in revised manuscript (See in dataset).

Specific comments

First, lots of data regarding the surface sediment in the Asian dust source region are missing. Only the dust samples, mostly from snow/ice, are insufficient to track the regional dust transport. Because a substantial amount of those dust from snow/ice is not in situ and probably long-distance transported, the regional distribution of the Nd isotope dataset mostly from those snow/ice dust cannot reflect the dust transport dynamic. In this regard, the box plot in Figure 3 divided by regional distribution can be biased, especially for those sand/soil samples.

Reply: We thank the reviewer for bringing this point forward. We agreed that the most of dust samples are from snow/ice, however, in some sub-regions, in our previous studies (Du et al., 2019a; Don et al., 2014) and the others (Wei et al., 2017; Dong et al., 2018), the local dust form the glaciers were also measured. Therefore, we added these data and redid this figure in the revised manuscript.

Yang et al. 2021 GCA (https://doi.org/10.1016/j.gca.2020.12.026) compiled a new Nd isotope distribution of surface sediment (desert, fluvial, moraine, loess, and soil samples) over east and central Asia from many previous studies and new data. Most of many previous studies are missing in the present manuscript, e.g., Blayney et al., 2019; Chang et al., 2000; Clift et al., 2017; Garzione et al., 2005; He et al., 2019; Li et al., 2009; Liu et al., 1994; Nakano et al., 2004; Rao et al., 2015; Wu et al., 2010; Zhao et al., 2014, 2015. In Yang et al. 2021. The authors can find those missing studies with detailed citation information and data in Yang et al.'s Table S1.

Reply: We thank the reviewer for bringing this important reference to our attention. We defined that the Third Pole covers the area of $40\,^\circ$ to $23\,^\circ$N and $106\,^\circ$ to $61\,^\circ$E (Li et al., 2020, https://doi.org/10.1175/BAMS-D-19-0280.2). Therefore, we mainly focused on Sr-Nd data from the snow or ice and referred the potential dust sources in last version. As you suggestions, we added and integrated some Sr-Nd data by Yang et al., (2021) into the entire Third Pole.

Second, many of those missing studies of Nd isotopes also reported Sr isotopes, which should be compiled in this study. Other studies for the Sr isotope of surface sediments

should also be cited. e.g., Jacobson, 2004 (https://doi.org/10.1029/2004GC000750)

Reply: Agreed. We have realized this problem and added Sr data as your suggestion.

Third, there are many Sr isotopic data regarding different phases of sediments (e.g., water-soluble salts, acetic acid leachate, acid-residue, etc., Honda et al., 2004 (doi: 10.1046/j.1365-3091.2003.00618.x); Yokoo et al., 2004 (doi:10.1016/j.chemgeo.2003.11.004); Nakano et al., 2005 (doi:10.1016/j.atmosenv.2005.05.050) and many other studies. If available, different phases of Sr isotopic data could also be compared.

Reply: It was not clear. Thanks for your these important references. Because the style of this Journal focuses on the dataset, therefore, we presented Sr-Nd isotopic data with different phases of sediments in dataset (See in the dataset).

Fourth, we know that there are many different units in the Himalayas with particular Sr-Nd isotopes (for example, see Jonell et al. 2018 https://doi.org/10.1016/j.chemgeo.2018.03.036). Those Sr-Nd isotopes of surface sediment should also be cited. If the authors think that the much variable Sr-Nd isotopes of source rock are not related to the present snow/ice dust samples, the authors should state the limit of the current compilation and explain the reason why the Himalaya dust yields a limited range of Sr-Nd isotopes within a region consisting of such variable source rocks. Nanga Parbat, the Lesser Himalaya, and the Indian craton yield the least radiogenic ε Nd values ranging from −26 to −23 (Ahmad et al., 2000; Clift et al., 2002; Robinson et al., 2001). The Greater Himalaya (ε Nd = −17 to −12) and Karakoram (ε Nd =−12 to −8) have intermediate values that are distinct enough for source discrimination (Ahmad et al., 2000; Crawford and Searle, 1992; Deniel et al., 1987; Inger and Harris, 1993; Parrish and Hodges, 1996; Schärer et al., 1990).

Reply: Thanks for your valuable reference. The Sr-Nd isotopes of source rock are not related to the present snow/ice dust samples, however, as you mentioned that the Himalayas is very important and complicated, in this study, the age of Sr-Nd data was limited to the Quaternary period. And the variable Sr-Nd isotopes of source rock may refer the geology in the three poles, Therefore, Sr-Nd data was not compiled from the rocks of three poles. And we did not further discuss this part in text.

Technical corrections

Line 76-70, there is another paper regarding Sr isotope as an eolian tracer Yang et al., 2017 (http://dx.doi.org/10.1016/j.epsl.2017.02.009).

Reply: Thanks, Done.

Sr-Nd correlation plot like figure 8 for the Antarctic could also be illustrated for those data in Asian dust sources and the Arctic.

Reply: We have fixed this.

---

## Author Comment (AC3)

Answer to reviewer comment 2

Review for Du et al. ESSD by Cécile Blanchet

Potsdam, April 19 th 2022

The authors present a data compilation for neodymium and strontium isotopes to trace dust deposition in the "three poles". In order to do that, the authors have assembled data from the literature as well as new measurements. I would recommend to reorganize the manuscript and better define the objectives of the study before this can be published. I have several comments, which I hope will help to improve the manuscript.

Reply: The authors greatly appreciate Prof. Cécile Blanchet for the very helpful and constructive comments in suggesting improvements on our first submission.

Please see our responses below, we have made major revisions to the manuscript, including significant additions for Sr-Nd data. We hope that this version addresses the comments of the referee.

Main comments:

1) I am not entirely sure what the main goal of the paper is: is it to determine where dust deposited on the three poles comes from? Is it to determine how the three poles act as sources and/or sinks through time? Is it to determine the role of local dust sources to the sinks on the three poles? On line 106, it is not clear which are the "questions" the authors are aiming to answer. Clarifying the aim will help to understand the rationale of compiling data from soil surfaces, glacier, snow, and marine sediment cores, but not using more global databases that would allow to determine the contribution from other potential source areas.

Reply: Thanks for your comments. We agreed that the main goal of the paper is not clear. First, the previous datasets (Blanchet et al., 2019; Robinson et al., 2021) had covered the most of regions in global scale. Therefore, the data was mainly compiled in the three poles. In this dataset, we defined the area of three poles based on the previous study (Li et al., 2021). Because the dust transport is complicated in Arctic, we did not use the PSAs at a global database. Second, for the remote three poles, which almost cover by the snow or ice. The long/short distance dust is transported and preserved into snow or ice by the atmospheric wet and dry depositions. Therefore, snow or ice from the three poles is the sink for the aeolian dust. And the deserts in arid areas of mid and low latitudes are the main source of dust. Third, as global warming, the exposed bedrock will be the dust source of snow and ice by physical or chemical weathering, which will become the new dust source. In addition, the sediment from Arctic Ocean and Southern Ocean will mix the aeolian dust and continental shelf material, which would be the sink. The age of samples was limited to the Quaternary period (2.5 Ma) because of the oldest ice core is about 800 000 years. The most of samples are collected from the surface deposits. Therefore, the aim of this manuscript is focused on the source and sink relationships between snow, ice and sediment records and potential source areas in this study. In the revised version, we reorganized the Introduction, the scientific topic of this work are further explained in this part.

2) If the aim of the paper is to describe the dataset, the authors need to spend more time explaining their strategy for data collection. Literature search? Use of previous compilations? Number of data from the literature versus samples measured and

published in this paper. Needs to be clarified, esp. in lines 175-190, where it is quite confusing (although it is clearer in the abstract). Please separate in the methods the literature search from the sample collection and measurements.

Reply: Yes, the describe dataset is the topic in this paper. In revised version, we recompiled much more Sr-Nd data and references. We presented the dataset information in Table 1. Because this strategy for data collection is the same with Blanchet (2019), we simplified this detail process, and as the examples, we try to list the patterns or characteristics in the three poles. The sample collection and measurements were separated in revised version.

3) Data file (excel spreadsheet): I struggle with the way the data are presented: I would recommend to use one template for all the data. In the attached spreadsheet, there are four different sheets with each a different template. For the sake of being able to compare the data with each other, please follow a strict (and similar) order and include "NA"s when an information is missing. If you want your data to be used by others, why not using the template that we published last year (version 3.0 of my database: https://dataservices.gfzpotsdam.de/panmetaworks/showshort.php?id=7124101c-d2a2-11eb-9603497c92695674), with perhaps modifying the "water depth (m)" column into an "elevation (m)" column (with changing signs whether it is altitude (+, masl) or bathymetry (-, mbsl)? Also please provide the locations in decimal degrees as it allows most programs to plot the data (it is more difficult with locations in minute/second). Also please provide the metadata in the downloadable spreadsheet.

Reply: Thanks for your concern. We are aware of it and used one template for all the

data as your version 3.0 database. The information of altitude or bathymetry and the locations were added in revised manuscript. The coordinates of locations were transferred in decimal degrees, and the metadata was also provided in revised manuscript.

4) Data visualisation: I think that one of the powerful possibilities of compiling data and attributing precise locations is to explore the spatial variability of certain variables by plotting isoscapes (instead of plotting numbers on locations like in Fig. 5). You're your case, it would be very interesting to compare the changes in Nd and Sr isotopes spatially in the three poles for the source material (soils) and for the sinks (snow, sediments). In general, the authors need to better separate sources and sinks throughout the manuscript.

Reply: Thanks, we agreed it. The Fig. 5 was replaced, however, we do not by plotting isoscapes for two reasons. First, few Sr-Nd data (including deep ice core) from snow or ice are collected from the Greenland ice sheet, which will result in the much uncertainty; Second, the Sr-Nd data from the Third Pole are multi-types, the geological unit is complicated, which are difficult to express with plotting isoscapes, and Sr-Nd data from the Antarctica are not uniformly distributed. As a case, we presented the Holocene samples for Antarctica with by inverse distance weighted interpolation using ArcGIS. We tried to identify the source or sink characteristics of Sr-Nd in dataset. However, for example, for loess samples from the deserts, Sr-Nd data may represent the source, but it may be sink in Chinese Loess Plateau because this material produced from the deserts and then deposited in this region. We try to separate sources and sinks in revised

manuscript as you suggestion.

5) Box and whiskers plots (Fig. 3 and 6): please indicate the number of samples used for each category. Please note that these analyses are best suited for >5 samples.

Reply: Thanks, we checked the box and whiskers plots in Figs. 3 and 6, the number of samples are all >5 samples, which were added in Figs.

6) In general, the isotopic signature needs to be discussed in terms of lithologicak context, which is seldom mentioned in the manuscript. Perhaps use geological maps to contextualise the varying signatures observed?

Reply: Agreed. Indeed, the lithological concentration for marine sediment can effect Nd values, we added the part of discussion about lithologicak context in lines 209-217. However, the geological conditions for the three poles are very complicated because of the areas of the three poles covering by snow or ice, and the resolutions of lithologicak data in Arctic Ocean and Southern Ocean are low, therefore, we did not contextualise it into the geological maps.

7) There are way too many abbreviations and I must say, I got lost. Please use abbreviations only when necessary but otherwise use the full names.

Reply: Sorry, we checked and corrected them in revised manuscript.

Minor comments:

L. 29-30: "recognized and introduced": what do you mean here?

Reply: We revised these word in revised manuscript.

L. 115: what is "data augmentation"?

Reply: We deleted it.

L. 122-123: it might be useful to clarify the expression "three poles" for the readers who might not be familiar with it. I knew the expression "third pole" but not "three poles". In lines124-125, you cite Australia, Southern south America, Southern Africa and New Zealand: are these regions part of the three poles? I thought the third pole refers to the Himalayas? Please clarify.

Reply: Thanks, we gave the "three poles" areas in Fig. 1. The areas of Australia, South America, Southern Africa and New Zealand are not the parts of the three poles. We collected the Sr-Nd data from the Himalayas and added it.

L. 126: I don't think that the abbreviation TP has been introduced before.

Reply: Changed.

L. 131: Please explain what a cryoconite sample is.

Reply: Explained it in lines 117-119.

l. 182: Very pleased that you used the scheme I developed: it will be very useful to compare and compile datasets!

Reply: We used the template as you developed it.

L. 218-219: I do not understand this sentence. The "acid leaching method" used is not given in the spreadsheets.

Reply: As you know, the gran sizes and acid leaching can effect Sr and Nd isotopic ration, therefore, we attempts to build a database that includes the different gran sizes and acid leaching methods (See dataset).

l. 220: "This feature validates" is a strange formulation, please clarify.

Reply: We deleted it for avoiding the misinterpretation.

L. 221-222: how did you determine PSAs exactly? Based on which criteria?

Reply: We determined the PSAs based on two criteria. First, the geographic location of deserts usually can be divided into the different geologic units. Such as, the arid regions from the western China, Sahara, Australian, South Africa and South America. In general, the PSAs mostly distributed in arid regions of low-mid latitudes in the northern Hemisphere. Second, a number of previous studies have identified the PSAs based on Sr-Nd data or dust transport model at a global scale (Chen et al., 2007; Li et al., 2008; Du et al., 2019; Zwaaftink et al., 2016; Dong et al., 2021). Therefore, Sr-Nd data will further demonstrate these identified PSAs, and try to link the relationship between PSAs and glaciers, and Sr-Nd data also can provide the new information for further finding the others possible PSAs.

L. 224-253: are these six regions the PSAs? How were they determined?

Reply: Yes. In general, these six region were determined by the high mountains (> 4000 m msl). Such as, Tienshan Mountain, Kunlong Mountain, Qilian Mountain, Hengduan Mountain, Himalayas and interior Tibet Plateau. In revised manuscript, we plotted the Mountains in Fig. 2. Sr-Nd data in each PSAs were the unique characteristics.

L. 301-306: belong to the methods. This happens repeatedly throughout the manuscript. Please all technical considerations should be put in the methods. However, the authors are often describing how other teams have sampled, which seems not really necessary. It might be advisable to the authors to edit the paper: there are often redundancies or unclear sentences, which hinder a smooth reading and a comprehension of the paper.

Reply: Thank you for your suggestion. These sentences were deleted. We also reedit

L. 374-376: I do not understand this sentence. I cannot see any homogeneity/heterogeneity from Fig. 6. Please clarify or modify Fig. 6.

Reply: Sorry, we explain it in revised manuscript.

L. 377. Antarctica: there are a lot of new data for Antarctica assembled in the paper of Robinson et al. (2021) in Chemical Geology (Open Access, https://doi.org/10.1016/j.chemgeo.2021.120119). The accompanying database version is V3.0: https://dataservices.gfz-potsdam.de/panmetaworks/showshort.php?id=7124101c-d2a2-11eb-9603-497c92695674)

Reply: Thanks, we updated some data from Robinson et al. (2021) for Antarctica.

L. 429-438: the separation in these different areas seems quite arbitrary. What is it based on? The authors mix sources and sink and it is not very helpful to understand the processes driving dust transportation I think.

Reply: There different areas were divided based on Sr-Nd data. We agreed that the sources and sinks are not clear. We gave some explanation in revised manuscript based on your suggestions, such as, lines 161-167, 238-241, 313-316.

L530-556. Conclusions

L. 540-543: I find this claim not really supported by the manuscript. The criteria were not used to determine which dust source was contributing to which sink. This needs to be better demonstrated, e.g., by separating sources and sink in the database or in the figures.

Reply: Thanks, we revised these sentences in lines 523-525, 533-535 and 538-540. As mentioned before, because the source or sink may be uncertain for the different region. We just separated sources and sink in the database.

L. 554. A PSA should be defined based on present-day knowledge of dust generation, e.g., based on remote sensing or geomorphological evidence. At minima it needs to be defined by the dust produced: grain-size distribution, mineralogy, isotope signature as well as by its geographical location.

Reply: Thanks, it is clear that PSAs in Antarctica are identified by remote sensing or geomorphological evidence. We revised this sentence in lines 541-545.

Figures:

I would encourage the authors to revise their figures to make them more informative.

Fig. 2: The numbers on the map (glaciers and deserts) need to be related to the names in a legend or in the caption.

Reply: We added these names of glacier in the caption.

Fig. 3: see my previous comment on box and whiskers plots.

Reply: Thanks, we checked it and added the explanations.

Fig. 4: There needs to be a legend on this map to help the reader to identify the different sample sets.

Reply: Done.

Fig. 5: This figure would be much better is plotted as an isoscape (using interpolation between known values), and with maps for sources and maps for sinks. Actually, I think that Fig. 4 and 5 could be combined in isoscapes.

Reply: Yes, we agreed. However, as the readers and editors' concern, we should introduce the data characteristics. Besides, the sources or sinks are very different for the different age samples. Therefore, we just compared Sr-Nd data in 12 sub-regions with the modern samples (surface samples from sediment), which give the patterns of Sr-Nd in the Arctic.

Fig. 6. See Fig.3. I struggle with the abbreviations: please put the full names.

Reply: Sorry, added them.

Fig. 7: same as comment for Fig. 5.

Reply: Done.

Fig. 8: What are the A, B, C and D areas? Perhaps choose colour codes to distinguish the main areas?

Reply: We changed it with Fig. 7

Data: See main comment 3)

Reply: Thanks, we added in the major revisions.

---

## Author Comment (AC4)

**Answer to reviewer comment 3**

It's of great significance to compile the radiogenic isotope compositions data of Sr and Nd at the "three poles" for researchers to further understand and trace dust transportation. In this respect, the database is necessary and meet the demand of many readers.

We appreciate that the reviewer 3 finds our work interesting. We agreed with the reviewer's comments for improvement. We hope that this version addresses the suggestions and comments of the reviewer 3 to improve the smooth flow of the text. The following are some comments to improve this manuscript:

**Main comments:**

I suggest to add a paragraph before Line 55 to explain why 87Sr/86Sr and 143Nd/144Nd were chosen to trace dust sources. That means the tracing principle of radiogenic isotope compositions of Sr and Nd should be introduced here. There were a mass of Sr-Nd data measured in the "three poles" in different medium (e.g. rock, clay, sediments, dust...), why only data from sand, sediment, loess, Aeolian deposits and snow/ice were compiled here? Were they all connect to Aeolian dust and easy to trace the sources? Is this manuscript only focus on "Cryospheric science"? There should be some words to explain these choice.

Reply: We explained why 87Sr/86Sr and 143Nd/144Nd ratios were chose to trace the dust sources in lines 50-56. We added Sr-Nd data from rock, sediments, dust and others from the Third Pole in revised manuscript. Of course, we concerned the dust in snow/ice, therefore, we gave the examples about dust tracing in snow or ice in the three poles. Some conclusions in the manuscript needs more robust evidence: L. 362-364: "As shown in Fig. 6, the lowest  $\varepsilon$  Nd values were observed along the ice-free periphery of the GrIS and SV; therefore, these ice-free regions are potential dust sources for natural dust in the Arctic Ocean." Do same/similar values of dust mean the same sources? It's better for the authors to give some robust evidence from modern or ancient atmosphere circulation pattern to explain the transport pathway.

Reply: Thanks for your suggestion, we agreed with you. In general, the same or close Sr-Nd values indicate the similar or same sources, but it need consider the other factors. As the reviewer of C cile Blanchet's suggestion, we reorganized the manuscript structure for describing the data. The scientific aspects were simplified, because the much scientific interpretations may be outside the scope of data article. Therefore, we tried to introduce the data characteristics and present the some cases in revised manuscript.

As the author mentioned in L.212-219, there exist "grain size effect" and "acid leaching method effect" that influence the Sr isotopic composition data, maybe there is also "altitude effect" on the glaciers, then the questions arise : How to avoid above "effects" when compare different data obtained from different grain sizes, different leaching methods or different altitudes?

Reply: The previous studies had demonstrated that the "grain size effect" and "acid leaching method can significantly influence the Sr isotopic ratios. In general, we should choose the similar or same grain size and acid leaching method when compared these data. The Sr isotopic ratios were controlled by the mineral components in solid state, which is different with the nonmetallic isotopes as the oxygen or hydrogen isotopes in liquid state. As your question, in fact, the "grain size effect" is similar with the phenomenon of "altitude effect". For example, the much finer dust particle will transport and deposit into the higher altitude region, while the coarse dust particle will transport and deposit into the lower altitude region. Because the "grain size effect" was widely used in geochemical field, therefore, we do not change it in revised manuscript. However, we discussed this effect (the different grain size and acid leaching method) in part 3.1 in revised manuscript. The Sr-Nd data of grain size and measurement methods were also added in dataset.

I recommend to reorganize table.2,3,4 with two comparison columns of "sink" and "source" radiogenic isotope compositions data of Sr and Nd. That needs more data from potential source areas to be collected in the database.

Reply: Thanks your suggestions, Sr-Nd data were marked the possible "sink" and "source" in the dataset. However, it is note that the possible "sink" and "source" may be changed because of the different depositional environments. The much more data was added in revised dataset, and we try to reorganize tables. 2, 3, 4 based on the updated data.

**Minor comments:**

L.99: "and much is still known about the cycle in the SH"ï¼ the "known" should be "unknown"?

**Reply: Thanks, we corrected it.**

L.161-162: "with the unit of at revolutions per minute (rpm)", a word was missed here.

Reply: The sentence was reorganized and deleted in revised manuscript.

L.194 and L.197: 144Nd/146Nd should be "143Nd/144Nd"?

Reply: Sorry, we corrected them in revised manuscript.

L.244-245: "with  $\varepsilon$  Nd (0) values from 0.712349 to 0.73211 and  $^{87}$ Sr/ $^{86}$ Sr values from

-15.7 244 to -7.0"? The position of " $\epsilon_{Nd}(0)$ " and "87Sr/86Sr" should be reversed.

Reply: Thanks for your reminder, we corrected them in revised manuscript.

Fig. 6: "CAA" at X axis should be "MCAA".

Reply: This has been fixed.

Table 3: "CAA" in the second column should be "MCAA"

Reply: Done, MCAA is not right, we kept the CAA in revised manuscript.

Fig. 6: What are the red and green rectangles mean?

Reply: The red and green rectangles mean the average values, we further added the referred explanations in lines 300-304.

L. 279-280: "Sand samples from PSAs (East Asian and Saharan deserts) are also collected." I could not find any data from Saharan deserts either in the manuscript nor the database.

Reply: Thanks. These data had been included in previous datasets (Blanchet et al., 2019; Robinson et al., 2021), we focused on the Sr-Nd data from the three poles, therefore, these data from Saharan deserts did not include in this dataset.

Fig.7: "Surface snow or snowpit samples are represented by purple solid circles, ice core samples represent blue rectangles and samples represent red solid circles". Where is blue rectangles? What samples represent red solid circles?

Reply: Sorry, the information of these symbols are wrong, we rephrased this sentence in revised Fig. 7.

Fig.7: "The dust transport paths are marked with yellow arrows." Do the dust transport paths here connect to the dust "sinks" of the Antarctic ice sheet?

Reply: Yes. These arrows indicated the possible dust "sinks". Because the dust sources are relative complicated for the entire Antarctic ice sheet. The much detail transport information cannot be described in here because of its complicated routines and different media, we added the relevant references for further explaining the routines (Gaiero, 2007. doi:10.1029/2007GL030520, 2007; Shao et al., 2010, doi:10.1016/j.aeolia.2011.02.001; Gili et al., 2022, https://doi.org/10.1038/s43247-022-00464-z). We try to mark the possible source or sink in this dataset. And we explained this phenomenon with Fig. 7, which it clearly showed the dust transport paths based on Sr-Nd data.

Fig. 8: What did the letter "A, B, C, D" mean? Do they refer to the four regions described in L. 429-438?

Reply: Yes, "A, B, C, D" mean four distinctly different Sr-Nd regions, we updated and deleted it.

---

## Referee Report (RR1)

Review for Du et al., revised version AR1
By Cécile Blanchet (GFZ Potsdam), 21 September 2022

Thanks to the authors for the extensive corrections they have performed on the manuscript and their detailed responses. I have only a few minor corrections before this paper can be published.

- The goals of the paper are clearer.
- The complexity of source versus sink determination in the third pole is well explained.
- The three poles are better defined in the text and Fig. 1 is very helpful.
- The figures have been largely improved, with number of data analysed for each box plot (Fig. 3 and 6), proper legends (Fig. 4 and 7) and isoscapes (Fig. 8).

I quickly checked the attached dataset, which is a lot clearer and better organised than the previous version. However, a **metadata description** is still missing, indicating what kind of information is given in the columns and the different spreadsheets. It is an important part of the dataset that help users to select and reuse the data.

L. 111: (…) and literature from the three poles (which refers to the high mountainous regions in Asia, the Arctic and Antarctica).

L. 166: what is the "pan-Third pole"?

Section 3.1: I am not sure how much of this section is really useful for the paper. All considerations on paleoceanography seems a bit out of scope.

---

## Author Response (AR2)

We thank the editors and reviewers for their very thorough and insightful comments. These comments greatly helped us to clarify the scope of this manuscript and to emphasize our key points throughout. We have studied comments carefully and have made correction which we hope meet with approval.

Referee 1

My major concerns are well addressed in the revised version. However, I find that there are many minor issues (grammar, vague wording, etc.), especially in those revised parts (highlighted in the tracked version) that should be carefully corrected. I list some of them below.

Reply: We would like to thank the referee 1 for his thorough evaluations with constructive comments that certainly will improve the manuscript. In the following, we will address the referee 1 comments point by point. We mark black the comments given by the referee, give our answers and comments in blue.

L122 one million years?

Reply: Thanks, we corrected it.

L166 Pan-Third pole?

Reply: Yes, because the locations of these data are beyond the area of Third Pole. The most of studies named these regions as Pan-Third pole.

L172-174 rewrite, poor grammar.

Reply: Thank you, done.

L174-176 rewrite, poor grammar

Reply: We rewrote it.

L199 remove "slightly", the grain-size effect is much more significant for 87Sr/86Sr indeed.

Reply: Done.

L208 It is unclear the meaning of "enrichment". Do you mean the radiogenic isotope enrichment, which means a higher 87Sr/86Sr ratio and a 143Nd/144Nd ratio? Please specify it.

Reply: We added it in line 206.

L203 You need to explain the carbonate effect because you have mentioned the grain-size effect before but never mentioned carbonate effect.

Reply: Thanks, we deleted "the carbonate effect" in revised manuscript.

L204-205 I guess you mean" exhibits less grain-size-dependent variability…"

Reply: Yes, we revised it in line 202.

L210 no than? Less than? Rewrite.

Reply: Thanks, it is corrected in line 208.

L219 such a linear relationship from a reference or this paper?

Reply: We added the reference in revised manuscript.

L225 before what ? Do you mean the $^{87}Sr/^{86}Sr$ ratio of acid residue is higher than the bulk sample before acid leaching? Please specify it.

Reply: Yes, $^{87}Sr/^{86}Sr$ ratio of acid residue is higher than without acid treatment or bulk sample. We added it in revised manuscript.

L350 I am not sure whether it is wise to use the abbreviation PSA to show potential source area, also at L430, L440. Two many abbreviations in this paper always confused me.

Reply: Thanks, we still used the abbreviation PSA because there are about 20 abbreviations. This abbreviation was used in mediums of snow or ice.

L391 what is the meaning "for bulk"?

Reply: We added the explanation in line 389.

L431 the sample number is >2

Reply: Thanks, done.

L436 Sr and Nd isotopic contours

Reply: Added it.

L444 I am not sure the meaning of "many more sample numbers"

Reply: Corrected it.

L529 poor grammar, rewrite.

Reply: Done.

L538 Sr-Nd isotopic data

Reply: Thanks, added it.

Table 1 Tibet Plateau to Tibetan Plateau

Reply: Thank you, done.

Table 2 Tiean Shan to Tien Shan

Reply: Thanks. We changed it.

Table 2 Kulun to Kunlun?

Reply: Kunlun. Sorry, we corrected it in revised manuscript.

Fig. 2 You use Tianshan in the figure and caption but Tien Shan in the Table.

Reply: We used the Tien Shan in all manuscript.

Fig. 2 The is not easy to read with such a colourful background.

Reply: We changed the colour of the numbers in Fig.2 in revised manuscript.

By **Cécile Blanchet** (GFZ Potsdam), 21 September 2022

Thanks to the authors for the extensive corrections they have performed on the manuscript and their detailed responses. I have only a few minor corrections before this paper can be published.

• The goals of the paper are clearer.

• The complexity of source versus sink determination in the third pole is well explained.

• The three poles are better defined in the text and Fig. 1 is very helpful.

• The figures have been largely improved, with number of data analysed for each box plot (Fig. 3 and 6), proper legends (Fig. 4 and 7) and isoscapes (Fig. 8).

Reply: We thank Prof. Cécile Blanchet for their careful and thorough review of the manuscript. Below we copy the referee comments in black and write our responses in blue.

I quickly checked the attached dataset, which is a lot clearer and better organised than the previous version. However, a metadata description is still missing, indicating what kind of information is given in the columns and the different spreadsheets. It is an important part of the dataset that help users to select and reuse the data.

Reply: Thanks for your suggestion, we added a metadata description in revised dataset.

L. 111: (…) and literature from the three poles (which refers to the high mountainous regions in Asia, the Arctic and Antarctica).

Reply: Thanks, added it.

L. 166: what is the "pan-Third pole"?

Reply: The Pan-Third Pole included Tibetan Plateau, Pamir, Hindu Kush, Tianshan,

Iranian Plateau, Caucasus, Carpathians, as well as the surrounding deserts and coasts (2018AGUFMGC51B..01Y). Because the Sr-Nd data is also collected from this region, therefore, we used the name in dataset. We added the explanation in revised manuscript.

Section 3.1: I am not sure how much of this section is really useful for the paper. All considerations on paleoceanography seems a bit out of scope.

Reply: We discussed it again, and think that Section 3.1 is needed, because it presented the different mediums, which had the different Sr-Nd values. Therefore, we keep these data in dataset. However, we agreed that the paleoceanography indeed seems a bit out of scope, we deleted some sentences.